# Histological Analysis of Gonadal Ridge Development and Sex Differentiation of Gonads in Three Gecko Species

**DOI:** 10.3390/biology13010007

**Published:** 2023-12-22

**Authors:** Izabela Rams-Pociecha, Paulina C. Mizia, Rafal P. Piprek

**Affiliations:** 1Department of Comparative Anatomy, Institute of Zoology and Biomedical Research, Jagiellonian University, 30-387 Krakow, Poland; iza.rams@doctoral.uj.edu.pl (I.R.-P.); paulinamizia@doctoral.uj.edu.pl (P.C.M.); 2Doctoral School of Exact and Natural Sciences, Jagiellonian University, 30-348 Krakow, Poland

**Keywords:** gonad, gonadal ridges, ovary, testis, development, sex differentiation, gecko, lizard

## Abstract

**Simple Summary:**

Gonads are organs such as testes and ovaries. They are organs with completely different structures which arise in development from common sexually undifferentiated primordia. The development of gonads has been well-studied only in some vertebrates, and lizards remain one of the groups with the least-explored gonadal development. In this study, we analyzed changes in the structure of developing gonads in three species of geckos from the formation of gonadal ridges, the primordia of gonads, through the sexual differentiation of gonads, to the formation of testes and ovaries. We showed that the earliest manifestation of the sexual differentiation of gonads is the differentiation in the thickness of the gonadal cortex and the number of cortical germ cells. The development of gecko gonads was characterized by a different tendency of the ovarian medulla to form cords. The ovarian cortex was thickened at the pole of the gonad and thin near the mesogonium. In differentiating testes, lumens formed very early. This study indicated a number of unique features in the development of geckos, as well as a number of common features in the development of their gonads and other vertebrates.

**Abstract:**

Reptiles constitute a highly diverse group of vertebrates, with their evolutionary lineages having diverged relatively early. The types of sex determination exemplify the diversity of reptiles; however, there are limited data regarding the gonadal development in squamate reptiles. Geckos constitute a group that is increasingly used in research and that serves as a potential reptilian model organism. The aim of this study was to trace the changes in the structure of developing gonads in the embryos of three gecko species: the crested gecko, leopard gecko, and mourning gecko. These species represent different families of the Gekkota infraorder and exhibit different types of sex determination. Gonadal development was examined from the formation of the earliest gonadal ridges through the development of undifferentiated gonadal structures, sex differentiation of gonads, and the formation of testicular and ovarian structures. The study showed that the gonadal primordia of these three gecko species formed on the most dorsally located surface of the dorsal mesentery, and both the coelomic epithelium and the nephric mesenchyme contributed to their development. As in other reptile species, primordial germ cells settled in the gonadal ridges, and the undifferentiated gonad was composed of a cortex and a medulla. Ovarian differentiation started with the thickening of the gonadal cortex and proliferation of germ cells in this region. A characteristic feature of the developing gecko ovaries was the thickened crescent-shaped cortex on the medial and ventral surfaces of the ovaries. The ovarian medulla also grew and exhibited diverse tendencies to form cords. In the leopard gecko, advanced cord-like structures with lumens were observed in the ovaries, which were not seen in the crested gecko. Testicular differentiation was characterized by cortical thinning and the disappearance of germ cells in this region. In the medulla, the development of distinct cords with early lumen formation was noted. A characteristic feature of embryonic gonads was their growth in a horizontal plane. In this study, gonadal development was characterized by several features that are shared by geckos and other reptiles, along with features that are specific only to geckos.

## 1. Introduction

All vertebrates share some common characteristics in gonadal development. For example, the primordia of the gonads, known as gonadal ridges or genital ridges, develop as a thickening of the coelomic epithelium on the ventral surface of the mesonephros, which is the embryonic kidney [1]. Primordial germ cells (PGCs) colonize the gonadal ridges and settle within the epithelium. They constitute a cell lineage that gives rise to sperm or oocytes in mature gonads. Moreover, they always originate in the regions of the embryo distant from the gonads and migrate through the dorsal mesentery, on which the intestine is suspended, to reach the developing gonadal primordia. Cells from the mesonephric mesenchyme also colonize the gonadal primordia. As a result, sexually undifferentiated gonads are formed. They are bipotential, which means that they have the capacity to give rise to testes or ovaries. The somatic cells of undifferentiated gonads express genes involved in sex determination. The process of sex determination is followed by that of gonadal sex differentiation, during which differences between ovaries and testes emerge. In general descriptions, it is often stated that the undifferentiated gonads consist of a cortex and a medulla: the cortex gives rise to the ovary, and the medulla—to the testis [2]. However, this is an oversimplification. A large portion of the ovary also develops from the medulla, while the cortex in the testis forms the epithelium covering the male gonad. Within the ovarian cortex, germline nests develop, which then fragment into ovarian follicles. In the testicular medulla, on the other hand, testis cords develop and transform into seminiferous tubules.

In addition to many similarities, there are also differences and modifications in gonadogenesis among vertebrates, even within a single group [3]. For example, in teleosts and mice, distinction between the cortex and medulla during development is not observable. In the ovaries of amphibians, the medulla transforms into a thin epithelium lining the cavity. In some teleosts and frogs, such as the zebrafish and several *Rana* species, testes undergo an initial phase of ovarian development. In birds, there is a clear asymmetry in size and structure, especially in the ovaries. When searching for gonads with the most primitive characteristics, reptiles serve as an example. Their gonadogenesis appears to be the least modified, although this area remains poorly studied. Little is known about the diversity of gonadogenesis among the different groups of reptiles. Studies on gonadal development in reptiles can provide insights into the evolutionary basis for the modified gonads of birds and mammals.

Sex determination is a much more diverse and evolutionarily plastic process compared with the structural features of the developing gonads. In most vertebrates, sex is genetically determined. However, in many species, sex is determined also by environmental factors. For example, temperature-dependent sex determination is characteristic for many reptiles, including all crocodile species, most turtles, and numerous lizards [4]. Many species have genetic sex determination, but the temperature can still influence sex. This phenomenon is known as temperature effects. It remains unknown whether these diverse mechanisms of sex determination in reptiles are somehow related to the structural diversity of their gonads.

The molecular and cellular processes of gonadal development in reptiles have been most extensively studied in turtles. In the olive ridley sea turtle (*Lepidochelys olivacea*), detailed electron microscopy studies showed structural changes that occur in the gonads of embryos incubated at different temperatures [5]. Males were shown to develop at a temperature of 27 °C, indicating a male-producing temperature (MPT), while females were shown to develop at 31 °C, indicating a female-producing temperature (FPT). Subsequent studies on this species revealed that the temperature influences sex only during a specific seven-day period known as the thermosensitive period [6]. Furthermore, exogenous estradiol leads to gonadal feminization, resulting in the development of ovarian characteristics in the gonadal structure despite being incubated at MPT. It was shown that under MPT, medullary cords develop with the expression of the *SOX9* gene involved in the development of vertebrate testes [7]. More advanced studies on the same species of turtle demonstrated the effects of exogenous estradiol on temperature-dependent sex determination [8] and cell proliferation in medullary cords [9].

The cellular and molecular aspects of gonadal development were also studied in the red-eared slider turtle (*Trachemys scripta*) [10]. This species has been particularly well-described in terms of the formation of medullary cords and the migration of cells to the developing gonads during the differentiation of testes and ovaries in embryos incubated at 26 °C (MPT) and 31 °C (FPT), respectively. Gonadal development was also studied in the American alligator (*Alligator mississippiensis*) [11,12]. Using electron microscopy, the authors described the formation of testicular and ovarian structures during the process of gonadal sex differentiation in embryos incubated at 33 °C (MPT) and 30 °C (FPT), respectively.

Squamate reptiles remain the least studied group in terms of gonadal development, even though they constitute the largest and most diverse order of contemporary reptiles. It comprises 11,549 species (http://www.reptile-database.org/db-info/SpeciesStat.html (accessed on December 2022)), which accounts for 96.7% of the currently known reptile species. Among squamates, experimental studies indicated the disruptive effect of atrazine on gonadal development in the skink *Niveoscincus metallicus* [13]. Some structural data on gonadal development are known for the skink *Niveoscincus ocellatus*, the anguid lizard *Barisia imbricata*, the agamas *Pogona vitticeps* and *Calotes versicolor*, and the phrynosomatid lizard *Sceloporus aeneus* [14,15,16,17].

Gekkota constitutes the most basal and one of the largest groups of lizards [18]. Therefore, research on this infraorder is significant for evolutionary reasons. Among reptiles, there is currently no single species to be used as a model species in research. Geckos possess characteristics that could make them widely useable in studies. For example, they are small, easy to keep in captivity, and they reproduce readily. Additionally, they exhibit various types of sex determination, which makes them attractive for research on gonadal development. Unfortunately, there are no histological data available on the gonadal development of the gecko. Several studies focusing on gene expression during gonadal development have been conducted in the leopard gecko [19,20].

The aim of this study was to investigate the structural changes that occur during the early stages of gonadal development, including the earliest formation of the gonadal ridges, the structure of undifferentiated gonads, the sex differentiation of gonads, and the structure of differentiated testes and ovaries in three gecko species representing different families and different types of sex determination (Figure 1A). Comparative studies such as this could reveal variations in gonadal development among the Gekkota as well as identify any common features of gonadal development in these lizard species. This seems to have important implications, because among the geckos studied in this study, there might be a species that holds the potential to become a model organism for future studies on lizard development. 

The first studied species, the crested gecko (*Correlophus ciliatus*), represents the Diplodactylidae family and is native to southern New Caledonia. It exhibits genetic sex determination with temperature effects and has heterogametic females (ZZ/ZW) [18]. The second species, the leopard gecko (*Eublepharis macularius*), is native to the grasslands and deserts of Afghanistan, Iran, Pakistan, India, and Nepal. It belongs to the Eublepharidae family, which, unlike the other geckos, lacks adhesive toepads and has movable eyelids. The leopard gecko has typical temperature-dependent sex determination [21]. Females predominantly develop when eggs are incubated at 26 °C, 30 °C, and 34 °C (FPT). However, males predominantly develop when eggs are incubated at 32.5 °C (MPT). The third species, the mourning gecko (*Lepidodactylus lugubris*), represents the Gekkonidae family. It is native to Southeast Asia, Indonesia, Oceania, and most islands of the Pacific, and has also been introduced to South America. This species is parthenogenetic, with occasional emergence of phenotypic males (approximately 1 per 400 individuals) that have impaired spermatogenesis [22].

## 2. Materials and Methods

### 2.1. Animals

The design of the study, involving the selection of appropriate days of sampling, was based on a previous publication describing gecko development [23]. We selected incubation days to isolate embryos from the formation of genital ridges, through the development of undifferentiated gonads, sexual differentiation of gonads, until the development of differentiated testes and ovaries (Figure 1B). The eggs of the crested gecko (*Correlophus ciliatus* Guichenot, 1866), the leopard gecko (*Eublepharis macularius* Blyth, 1854), and the mourning gecko (*Lepidodactylus lugubris* Duméril & Bibron, 1836) were obtained from private breeders in Kraków (Lesser Poland Voivodeship, Poland). Eggs were collected on the day they were laid and incubated in a wet vermiculite mixture (200 g of vermiculite and 160 mL of water) in closed boxes to prevent the eggs from drying out. The eggs were incubated at different temperatures: *C. cilliatus* at 27 °C, *E. macularius* at 26 °C (to produce females) and 32 °C (to produce males), and *L. lugubris* at 25 °C. The number of embryos tested at the consecutive stages is presented in Appendix A. This study was conducted in accordance with the Act on the Protection of Animals Used for Scientific or Educational Purposes (Dziennik Ustaw 2015 poz. 266 [Journal of Laws from 2015, item 266]).

### 2.2. Histological Method

Embryos dissected from eggs were staged according to Wise and colleagues [23], fixed in the Bouin’s solution overnight, dehydrated, and embedded in paraffin (Paraplast P3683, Sigma, Livonia, MI, USA). Samples were serially sectioned at 6 μm and stained with Harris-modified hematoxylin and picroaniline according to the Dubreuil’s procedure [24,25]. This method specifically stains the extracellular substance, thus exposing the internal structure of the organs. It also differentiates germ cells from somatic cells. The images were taken with a Nikon Eclipse E600 light microscope.

### 2.3. Immunohistochemistry

Samples fixed in Bouin solution and embedded in paraffin were serially sectioned at 6 μm. Sections were deparaffinated, rehydrated and heat-induced epitope retrieval was performed using sodium citrate buffer (10 mM sodium citrate, 0.05% Tween-20, pH 6) at 95 °C for 20 min. The procedure was conducted using the HRP/DAB Detection IHC Kit (ab64264, Abcam, Cambridge, UK). Incubation with the following primary antibodies was performed at 4 °C overnight: anti-E-cadherin (ab152102, Abcam), anti-laminin (L9393, Sigma), anti-PCNA (HPA030521, Sigma). Mayer’s hematoxylin was used as a counterstain. The images were taken with a Nikon Eclipse E600 light microscope (Amstelveen, The Netherlands).

### 2.4. Statistical Analysis

The thickness of the ovarian and testicular cortex, and ovarian and testicular medulla, and the number of germ cells were measured on three cross-sections at the widest point of three individuals at the same stage. The measurements of male and female gonads were compared using the χ^2^ test. The results of these analyses and the number of individuals analyzed are presented in Appendix A. The analyses were performed using the Statistica software package (version 12.0 StatSoft PL).

## 3. Results

### 3.1. Gonadal Development in the Crested Gecko (Correlophus ciliatus)

#### 3.1.1. Formation of the Gonadal Ridges

Gonadal ridges in the crested gecko formed at stage 29 (S29), that is, day 5 (D5) of egg incubation after laying (Figure 1B and Figure 2A; gonadal ridges are encircled). The earliest sign of the gonadal formation was the appearance of two bulges of the thickened coelomic epithelium located at both sides of the dorsal mesentery (Figure 2A–C; ce indicates coelomic epithelium; dm, dorsal mesentery; gr, gonadal ridges). These ridges formed on the dorsal end of the dorsal mesentery, rather than on the surface of the mesonephros (Figure 2A–C; ms indicates mesonephros). The coelomic epithelium in the gonadal ridge was more than twice as thick as the coelomic epithelium of the dorsal mesentery (mean thickness, 8.78 ± 2.74 μm and 4.13 ± 1.66 μm, respectively; Appendix A). The cells of the coelomic epithelium in the gonadal ridges were tightly packed. The ridges formed by the transformation of a thin simple squamous epithelium into a thickened cuboidal or columnar epithelium, are mainly simple and locally stratified. The space located inwards from the coelomic epithelium was filled with mesenchymal cells (Figure 2A–C; mc, mesenchymal cells). These mesenchymal cells of the dorsal mesentery accumulated adjacent to the epithelium of the gonadal ridges. The thickened coelomic epithelium constituted the primitive gonadal cortex.

Soon after the beginning of gonadal ridge formation, the first germ cells appeared in the gonadal ridges (S30/D6) (Figure 2D; gc, germ cells). The germ cells were easy to recognize owing to their large size, spherical shape, pale cytoplasm, large, pale, and spherical nucleus with several nucleoli, blue-stained Balbiani body, and yolk platelets (Figure 2E, insert). Yolk in the germ cells disappeared soon after the completion of their migration at S30. The somatic cells could be easily distinguished from the germ cells because they were smaller and had small and dark nuclei. The first germ cells settled between epithelial cells of the primitive gonadal cortex and became enclosed by them. At the next stage (S31/D7), the germ cells were present not only in the gonadal cortex but also in the gonadal medulla (Figure 2E). Considering the gonadal medulla, as early as at S30, its cells gathered in groups divided by the extracellular matrix (indicated by black arrowhead in Figure 2D), which indicated the beginning of medullary cord formation (Figure 2D).

From the moment the gonadal ridges were formed (S30) and throughout the whole process undifferentiated gonadal development (S31 and S32), the gonads grew and became more prominent, visibly bulging into the coelomic cavity (Figure 2F,G). At S32/D8, the undifferentiated gonads were well developed with a thick cortex covering the medulla (Figure 2H; co, cortex; md, medulla; the dashed line shows the border between the cortex and medulla). The cortex had a form of a cuboidal and locally columnar epithelium, simple or stratified, lying on the basement membrane (Figure 2H; bm, basement membrane). The germ cells within the cortex were enclosed by epithelial cells, whereas those in the medulla were located in the medullary cords and were also enclosed by somatic cells (Figure 2H).

Immunohistochemical staining of the proliferation marker PCNA revealed a strong signal in all cells in undifferentiated gonads, including the somatic cells of the developing cortex, medulla, and germinal cells (Figure 3A). Immunostaining of E-cadherin showed that this protein is a marker for the developing gonadal cortex and developing medullary cords from the beginning of their formation. In early gonadal ridges, the surface epithelium was E-cadherin-positive (E-cadherin+), while within the gonad, clusters of E-cadherin+ cells were interspersed with E-cadherin-negative (E-cadherin–) cells (indicated by arrows in Figure 3B). In subsequent stages of development of undifferentiated gonadal, E-cadherin+ cords extended from the interior of the gonad to the wall of the nearest Bowman’s capsule (Bc on Figure 3C). These cords were separated from the E-cadherin+ cortex by E-cadherin– mesenchymal cells (gonadal stroma) (indicated by arrows in Figure 3C). In a well-developed undifferentiated gonad, immunostaining of E-cadherin highlighted a distinct cortex resembling pseudostratified columnar epithelium, medullary cords located in the center of the gonad and separated from the cortex by a layer of mesenchyme; germinal cells are situated in the cortical epithelium adjacent to the mesenchyme (Figure 3D). Immunostaining of laminin demonstrates the presence of this protein in the mesenchyme (gonadal stroma; arrows on Figure 3E), forming a layer between the gonadal cortex and the medullary cords. The space devoid of laminin indicates the site of invagination of the developing cord from the cortex (Figure 3F).

#### 3.1.2. Ovarian Differentiation and Development

At S32/D10, the first differences between the ovaries and testes were noted, marking the beginning of gonadal sex differentiation. In the differentiating ovaries, the division into the cortex and medulla separated by the basement membrane was maintained (Figure 1B and Figure 4A; bm, basement membrane separating the cortex, co, and medulla, md). Thus, the general gonadal structure established in the undifferentiated gonads was continued. Cortical thickness increased along with a rise in the number of somatic and germ cells in the cortex. The thickest part of the cortex was located on the gonad’s distal pole (opposite the mesogonium) (Figure 4A). The cortex of the ventral surface was of medium thickness; however, the cortex in the dorsal part of the ovary and the part nearest to the mesentery was thinner (compare co1, co2 and co3 in Figure 4A).

The developing ovaries grew, elongated, and became ovoid at the cross-section and bilaterally flattened (Figure 4B,C). Proliferating (PCNA+) cells were present in the cortex and medulla (Figure 4D). A layer of cells separating the cortex and medulla showed lower signal of E-cadherin, indicating that it is a distinct cell population, probably the stromal cells (arrow on Figure 4D). The direction of the gonadal growth was unique (different from that in other vertebrates) because the ovaries grew dorsomedially towards the dorsal mesentery. As a result, the gonads folded up. Due to the growth of the mesonephroi, the gonads were anchored at the ventromedial surface of these organs (Figure 4B).

In the medulla, the cord-like structures were present. E-cadherin immunostaining showed a presence of streams of E-cadherin+ somatic cells extending from the cortex to the Bowman’s capsule in the mesonephros (Figure 4E). These streams of cells constituted the primordia of medullary cords and gonadal efferent tract. A zone of E-cadherin– cells lay between the forming medullary cords and the cortex. This zone was rich in laminins (Figure 4F), indicating that it constituted the mesenchymal nature of the gonadal stroma.

The blue-stained streams of the extracellular matrix indicated the borders of the cords (black arrowheads in Figure 4A,C). Only singular germ cells were encountered in the ovarian medulla (Figure 4G,H; germ cells in the medulla were pointed by white arrowheads). Along with the ovarian development (analyzed stages S35 and S36; days D17, D19, D22, and D25), the growth of both the ovarian cortex and the ovarian medulla was noted. Importantly, the germ cells became more numerous in the cortex (Figure 4G–M; germ cells in the cortex are pointed by red arrowheads). The cortex acquired a crescent-shaped form, being the thickest in the region of the distal end (co1 in Figure 4I), farthest from the mesogonium, and in the ventral region (co2 in Figure 4I), while it was the thinnest in the dorsal region (co3 in Figure 4I).

The details of the ovary at S36/D25 are shown in Figure 4M and N. The characteristic shape of the cortex, along with its varying thickness, as seen in the previous stages, was visible. The ovarian cortex was strongly E-cadherin-positive (co on Figure 4N). Pale germ cells were located between columnar somatic cells of the cortex. The ovarian medulla constituted a voluminous portion of somatic cells with no evident cords. Only streaks that probably represent underdeveloped cords were revealed by E-cadherin immunolabeling. These streaks were separated by groups of E-cadherin-negative stromal cells (arrow on Figure 4N). No germ cells were detected in the ovarian medulla at this stage. The extragonadal somatic cells in a streak (asterisk on Figure 4N) located in the mesogonium and extending to the Bowman’s capsule showed stronger signal of E-cadherin compared to the medullary cells.

#### 3.1.3. Testicular Differentiation and Development

At S32/D10, the first subtle signs of testicular differentiation were visible (Figure 5A,B). These signs were as follows: (1) the formation of the cord structures in the gonadal medulla continued, indicating the formation of the testis cords; (2) these cords were more solid compared with the differentiating ovaries, and the somatic cells in the forming testis cords showed clear epithelial differentiation and were arranged in rows (Figure 5A,B; compare with Figure 4A); (3) a small lumen appeared inside the testis cords (asterisk in Figure 5A,B); (4) the number of germ cells in the medulla increased and was higher compared with that in the differentiating ovaries; (5) the thickness of the cortex decreased (dashed line on Figure 5A indicates the border between the thinning cortex and medulla); and (6) the number of germ cells in the cortex decreased. At the stage of sex differentiation, the mean thickness of the cortex in the differentiating testes was 15.6 ± 5.4, compared with 36.4 ± 6.8 in the differentiating ovaries (Appendix A). The mean number of germ cells in the cortex of the differentiating testes was 0.9 ± 0.9 per cross-section, compared with 3.3 ± 0.7 in the differentiating ovaries. Finally, the mean number of germ cells in the medulla of the differentiating testes was 2.8 ± 0.9 per cross-section, compared with 0.8 ± 0.8 in the differentiating ovaries.

PCNA+ proliferating cells were present in the interior of the differentiating testis, and not in the periphery as in the differentiating ovaries (Figure 5C). E-cadherin immunostaining showed a strong signal in the testis cords (Figure 5D). These cords were contiguous with the forming efferent tubules. Laminin immunostaining showed that gonadal stroma was dispersed in differentiating testes and did not form one layer between the cortex and medulla as in the ovaries (Figure 5E).

At S34/D15, the testicular structure could be easily recognized (Figure 5F–H). The cortex was already transformed into a thin epithelium covering the gonad (the epithelium contained no germ cells, or only singular germ cells were occasionally seen) (Figure 5G,H). In the interior of the testis, the testis cords with a characteristic structure were present. The lumen was found inside the cords, indicating the early transformation of the testis cords into the tubules (Figure 5G,H; asterisks indicate the lumen). The wall of the testis cords was built of differentiating Sertoli cells (Sc in Figure 5H) arranged in a simple columnar epithelium with basally located nuclei (Sn in Figure 5H). Germ cells (prospermatogonia [indicated by red arrowheads in Figure 5G,H]) were located between the Sertoli cells. The testis cords were surrounded by the basement membrane (bm in Figure 5H), and the forming interstitial tissue (i in Figure 5G,H) filled the space between the cords.

As in the ovaries, the testes grew dorsomedially towards the dorsal mesentery (Figure 5F). In contrast to the ovaries, both sheets of the mesogonium in the testes approached each other. As a result, the testes were suspended on thin mesogonium (mg in Figure 5F).

At the next stages (S35 and S36; D17, D22, and D25), the testes grew and became filled with numerous testis cords (Figure 5I–O). At S36/D25, the lumen of the testis cords became wider (asterisks in Figure 5M–O). A primordium of the efferent ducts was present in a form of cords of E-cadherin+ cells (asterisk on Figure 5K) reaching from the testis cords in the gonads to the nearest Bowman’s capsule (Bc on Figure 5K).

#### 3.1.4. Intersexual Gonads

The gonads of two individuals assessed in this study had intersexual features (Figure 6A,B). The persistence of a thick cortex with germ cells was characteristic for the ovary. On the other hand, the presence of well-developed testis cords containing the lumen and germ cells was characteristic for the testis (Figure 6A,B; l, lumen in the testis cords).

### 3.2. Gonadal Development in the Leopard Gecko (Eublepharis macularius)

#### 3.2.1. Formation of the Gonadal Ridges

In the leopard gecko, the gonadal ridges formed at S29, as in the crested gecko (Figure 1B and Figure 7A–C). It corresponded to day 5 or 8, depending on the temperature of incubation. The site of the gonadal ridge formation was the same as in the crested gecko—the primordia of the gonads appeared at the border between the dorsal edge of the dorsal mesentery and ventromedially to the mesonephroi. Moreover, the gonadal ridges in the leopard gecko also had a form of two bulges composed of (1) a primitive cortex derived from a thickened coelomic epithelium and containing germ cells; and (2) an accumulation of mesenchymal cells located inside the gonadal ridge (Figure 7D; gr, gonadal ridges).

As in the crested gecko, the beginning of gonadal ridge formation in the leopard gecko occurred through the transformation of a thin squamous coelomic epithelium (mean thickness, 2.70 ± 0.97 μm) into a thick cuboidal or columnar epithelium (mean thickness, 8.58 ± 3.03 μm; Appendix A). In the center of the gonad, in both species there was a cluster of somatic cells continuous with the mesonephric mesenchyme. Moreover, as in the crested gecko, the first germ cells in the leopard gecko were observed among the somatic cells of the primitive cortex. The germ cells had similar morphology to that described in the crested gecko (Figure 7D, insert).

At S31, the undifferentiated gonads were larger, elongated at the cross-section, and they visibly bulged into the coelomic cavity (Figure 7E,F). The undifferentiated gonads suspended on the ventromedial surface of the mesonephroi are shown in Figure 7E. The structure of these gonads was clearly divided into the peripheral cortex and centrally located medulla, separated by the basement membrane (Figure 7F; bm, basement membrane). Singular germ cells were visible in the cortex, but they were also occasionally encountered in the medulla.

At S32, the medullary cords formed in the gonadal medulla (Figure 7G). Singular germ cells were present in the medullary cords (marked by gc in Figure 7G). This is different from the crested gecko, in which no well-differentiated medullary cords formed in the undifferentiated gonads (compare Figure 2H and Figure 7G). In both gecko species, the cortex of the undifferentiated gonads at S32 was composed of columnar epithelial cells arranged into a multi-layered epithelium (compare the cortex marked by co in Figure 2H and Figure 7G).

E-cadherin immunostaining showed E-cadherin+ cortex and also E-cadherin+ cords in the gonadal center (Figure 7H,I). The medullary cords extended from the cortex to the mesonephros. E-cadherin– cells filled the space between the cortex and the medullary cords.

#### 3.2.2. Ovarian Differentiation and Development

Differentiating gonads at S34 exhibited differences in PCNA immunostaining, revealing distinct distributions of proliferating cells (Figure 8A,B). In differentiating ovaries, PCNA+ cells were present in the cortex, indicating a higher proliferation rate of cortical cells. Conversely, in differentiating testes, they were scattered in the center of the gonad, suggesting a more robust proliferation of medullary cells.

At S34, ovarian differentiation in the leopard gecko could be identified due to an increase in cortical thickness and germ cell count as compared with testicular differentiation (Figure 9A). The presence of a thick, columnar cortical epithelium which developed in the undifferentiated gonads was maintained in the developing ovaries. As in the crested gecko, the thickest cortex was noted at the ventromedial pole of the ovary (opposite to the mesogonium). However, the dorsal surface of the ovary and regions near the mesogonium were covered by a thin simple squamous epithelium with no germ cells. Therefore, the whole ovary was not covered by a thick cortex. 

Importantly, in the leopard gecko, the medulla of all the differentiating ovaries contained the medullary cords with a lumen inside (Figure 9A; the cord is encircled; lumen is indicated by l). This indicated a strong tendency of the gonadal medulla to testicular differentiation even in the differentiating ovaries. Nevertheless, the medullary cords in the ovaries were sterile, that is, they contained no germ cells. The wall of the medullary cords was formed by a simple columnar (locally cuboidal) epithelium. The space between the cortex and medullary cords was filled by stromal tissue with blood vessels.

During the subsequent stages (S35 and S37), the structure of the cortex did not change significantly (Figure 9B–F). Cortical thickness increased only slightly between S34 and S37 (mean thickness, 30.53 ± 2.32 μm and 38.85 ± 2.46 μm, respectively; Appendix A). Also, the lumen of the medullary cords slightly increased in size (mean diameter, 11.63 ± 1.67 μm at S35 vs. 15.45 ± 2.85 μm at S37).

In developing ovaries, E-cadherin immunostaining revealed E-cadherin+ cortex as well as E-cadherin+ medullary cords, separated by E-cadherin– stromal cells (black arrows on Figure 9G). The zone of E-cadherin– stromal cells contained laminin, indicating their mesenchymal nature (Figure 9H).

As in the crested gecko, the gonads in the leopard gecko grew horizontally and were located medially towards the mesentery or they even curled up and grew dorsally (Figure 9C).

#### 3.2.3. Testicular Differentiation and Development

At S32/D10, the gonads of embryos incubated at MPT had germ cells located in the gonadal medulla but no germ cells in the gonadal cortex (Figure 10A). This subtle feature indicated testicular differentiation. The cortex at this stage was still thick and composed of a columnar epithelium.

At S34, the structure of the testis was easily recognizable due to the lack of a thick cortex (Figure 10B,C). Thus, as in the crested gecko, the thick cortex of the undifferentiated gonads in the leopard gecko was transformed into a thin simple squamous epithelium covering the testes. This process was accompanied by the loss of cortical germ cells. Inside the developing testes, the medullary (testis) cords were present. The germ cells were located within the epithelium of the wall of the cords, and the lumen was visible within the cords. Beginning with this stage, the testes were bilaterally flattened.

At all stages assessed in this study, the developing testes in the leopard gecko were bilaterally flattened and oval at the cross-section. In contrast, the testes of the crested gecko were not flattened and were spherical at the cross-section (compare Figure 5L and Figure 10B). In both species, the testes were located horizontally.

At S37, the testis cords were more numerous and the lumen inside them was larger (Figure 10D; testis cords indicated by tc). The epithelium forming the wall of the testis cords in the leopard gecko was cuboidal and not columnar as in the crested gecko (compare Figure 5H and Figure 10D).

At S41, the testes were visibly larger but still flattened with a pointed tip (Figure 10E,F). The structure of the testis cords with a cuboidal epithelium formed by E-cadherin+ Sertoli cells with germ cells in between is shown in Figure 10G. E-cadherin– interstitial tissue (i on Figure 10G) filled the space between the testis cords, and the testis was covered by a thin and sterile simple squamous epithelium.

### 3.3. Gonadal Development in the Mourning Gecko (Lepidodactylus lugubris)

#### 3.3.1. Formation of the Gonadal Ridges

In the mourning gecko, the first sign of gonadal ridge formation was noted at S29 (Figure 1B). It was marked by the development of a thickened epithelium at both sides of the border between the dorsal edge of the mesentery and the mesonephroi (mean epithelial thickness in the gonadal ridge and dorsal mesentery, 9.82 ± 2.95 μm and 3.81 ± 1.01 μm, respectively; Appendix A) (Figure 11A–C; gonadal ridges are indicated by gr). At this stage, also the first germ cells were present in the gonadal ridges (Figure 11C; a germ cell is pointed by gc). The morphology of the germ cells was similar to that in the crested and leopard gecko. Yolk droplets were present when the germ cells invaded the gonadal ridges at S29, but the yolk disappeared soon after as in other geckos (Figure 11C, insert).

The undifferentiated gonads at S31 had a typical structure with a visible division into the cortex and medulla (Figure 11D; the border of both parts indicated by a dashed line). The cortex had a form of a cuboidal epithelium with anchored germ cells. The medullary cord is visible in the medulla (Figure 11D–F). This E-cadherin-positive cluster extended to the mesonephros (Figure 11E). A layer of E-cadherin-negative stromal cells (black arrows on Figure 11E) was present between the medulla and the cortex, both positive for E-cadherin.

#### 3.3.2. Ovarian Differentiation and Development 

At S32, the gonads in the mourning gecko had a structure comparable to the differentiating ovaries in the other geckos due to the presence of a thick cortex containing numerous germ cells (Figure 11F; the cortex is indicated by co). As in the other geckos, the cortex showed a gradient of thickness, and the thickest cortex (co in Figure 11F) was located at the ventromedial pole of the gonad located opposite the mesogonium (mg in Figure 11F). In this part of the cortex, numerous germ cells were present (red arrowhead points the germ cells in the cortex in Figure 11F). However, the cortex was thinner towards the mesogonium, and no germ cells were present. The basement membranes surrounding the clusters of cells in the medulla indicated the presence of the medullary cords (Figure 11F, basement membranes pointed by bm, and medulla by md). There was no lumen inside them at this stage. Single germ cells could be seen in the medullary cords (white arrowhead points the germ cells in the medulla in Figure 11F). 

A similar ovarian structure was noted at S34; however, the gonadal medulla was larger (Figure 11G). In some samples, the cortex was reduced with few or no germ cells, but the medullary cords were well developed, resembling testicular differentiation (Figure 11H,I; yellow dashed line points the border between the cortex and medulla). At S36, the first signs of lumen formation in the medullary cords were clearly seen (asterisk in Figure 11H,I). 

At S38, the developing ovaries had a thick multilayer cortex with numerous germ cells (Figure 11J–L). The germ cells were also present in the ovarian medulla (white arrowheads in Figure 11K,L). Contrary to the two other gecko species, the cortical epithelium in the mourning gecko did not have a columnar structure. In the ovarian center, the medullary cords were present with a lumen inside (asterisk in Figure 11K). However, in contrast to the ovaries in the leopard gecko, the wall of the medullary cords in the mourning gecko was not well organized and the borders of the cords were not clearly seen. Immunostaining showed the presence of the cortex with a strong signal of E-cadherin, forming a cluster at the gonadal pole (Figure 11M,N). Pale germ cells were visible between the cortical somatic cells. In the medulla E-cadherin+ cells formed groups dispersed between E-cadherin– stromal cells (black arrows on Figure 11M). This indicated the presence of the rudimentary medullary cords. In the center of some groups of E-cadherin+ medullary cells, irregular lumens were present (black asterisk on Figure 11N). This medullary structure resembled the medulla of the developing ovaries in the crested gecko.

## 4. Discussion

The analysis of the three gecko species revealed gecko-specific characteristics in their gonadal development as well as some characteristics that are shared with other reptiles. The changes in the gonad structure of the studied species of geckos are presented in Figure 12. The initial stages of gonadal ridge development seem to be similar in the three species. According to the literature, it is commonly believed that the gonadal primordia, known as the gonadal ridges, form on the ventral surface of the mesonephros. They emerge close to the dorsal end of the dorsal mesentery. A similar site of gonadal ridge development was described in other reptiles, such as the phrynosomatid lizard *S. aeneus* [16]. In the gecko species studied here, gonadal ridge formation was noted at S29. This is in line with the reports for the lizard *Sceloporus undulatus* [26]. Embryonic stage 29 can be easily identified due to the initiation of limb bud formation. At the same stage of somatic development, specifically the limb bud stage, gonadal ridges begin to form in the toad *Bombina variegata* (Gosner stage 25), the frog *Xenopus laevis* (Nieuwkoop-Faber stage 48), the chicken (HH stage 20), the mouse embryos at embryonic day E10.0, and even the zebrafish, where the development of paired fin buds coincides with the formation of the gonadal ridges (about 20–28 h post fertilization) [27,28,29,30,31,32]. This indicates that at such an early stage, the initiation of gonadal development is synchronized with the limb bud development.

The most important characteristic that is common for all the studied geckos is the fact that the earliest manifestation of gonadal ridge formation is the thickening of the coelomic epithelium. This was also described in birds and mammals [30,33]. This is particularly evident in amniotes, because they have relatively small cells that easily accumulate in cluster where primordial germ cells settle. The formation of a thickened epithelium is not evident in amphibians, where massive primordial germ cells settle in the gonadal ridge, and the epithelium surrounding these cells becomes flattened [27]. Differences in cell size between amphibians and amniotes lead to slightly different structures of their gonadal ridges.

Another process observed in the developing gonadal ridges is the accumulation of mesenchymal cells in the gonadal primordia. Mesenchymal cells accumulate on the opposite inner side of the basement membrane of the coelomic epithelium in the ridges. Thus, at an early stage of development, the division of the developing gonad into the cortical and medullary region becomes apparent. This division has been observed in all the investigated reptiles so far, and it is also visible in the gonads of birds as well as numerous mammals such as bovines, but not in mice, in which gonadal development is modified [1]. Importantly, in all examined amphibians and amniotes, the germ cells remain close to the surface of the undifferentiated gonad. They are present in the developing cortex but not in the medulla, where they appear later and at a lower number. It is likely that primordial germ cells initially settle within the thickened coelomic epithelium, and then some of them migrate to the medulla along with a wave of somatic cells, as demonstrated in the slider turtle *T. scripta* [10]. This indicates that at least some medullary cells in the gonad originate from the cortex rather than solely the from accumulating mesonephric mesenchyme. In this study, we have demonstrated that, in geckos, the medullary cords are connected to the cortex, indicating that they form as an outgrowth of the cortex, similar to what has been shown in *T. scripta* [10]. This indicate the origin of the medullary cords from the gonadal cortex.

The undifferentiated gonads of the three gecko species examined here, as well as of other reptiles, are composed of the cortex and medulla. The cortex is continuous with the coelomic epithelium and contains nested germ cells. Such a structure of the undifferentiated gonads appears to be universal among reptiles and was also described in the lizard *C. versicolor*, the anole lizard *N. ocellatus*, the iguanid lizard *S. undulatus*, as well as in the olive ridley sea turtle *L. olivacea*, the slider turtle *T. scripta*, the spiny softshell turtle *Apalone spinifera*, and the American alligator *A. mississippiensis* [12,14,15,26,34,35].

In this study, we noted a tendency for cord formation in the medulla of the undifferentiated gonads in the leopard and mourning geckos, but not in the crested gecko. Apparently, this process can be induced at different stages depending on the species. The presence of medullary cords can be observed in *C. versicolor*, *S. undulatus*, *S. aeneus*, *L. olivacea*, *T. scripta*, *A. spinifera*, and *A. mississippiensis* [6,12,14,16,26,34,35].

After the undifferentiated gonad stage, gonadogenesis occurs, which is sex differentiation where the first differences between the developing ovaries and testes become apparent. In the examined geckos, this occurred at S32. This stage can be easily identified when the edge of the autopodium in the limb buds becomes wavy, indicating the beginning of toe development. The sex differentiation of gonads occurs at the same stage also in *Xenopus*, chicken, and mouse (unpublished data). Therefore, the onset of gonadal sex differentiation, just as the formation of the gonadal ridges, is another event synchronized with the limb buds.

In numerous previous studies, S34 was considered to mark the onset of gonadal sex differentiation, because at this stage, it is easy to distinguish the testes from the ovaries [14,16]. Only very thorough analyses supported by measurements of gonadal cortex and medulla and germ cell number can indicate sex differences at earlier stages. The initial manifestations of gonadal sex differentiation are subtle and involve cortical development. In the differentiating ovaries, the cortex becomes thicker and the number of germ cells increases. On the other hand, in the differentiating testes, the cortex gradually becomes thinner and germ cells disappear. It is possible that genes involved in female sex determination enhance the proliferation of somatic and germ cells in the cortex, while genes involved in male sex determination inhibit their proliferation. Future studies should focus on the regulation of proliferation and apoptosis in gonads as a molecular mechanism of gonadal sex differentiation. In this study, we demonstrate that in the differentiating testis, proliferating cells are scattered within the gonad and are not present on the surface (cortex). In the differentiating ovary, on the other hand, proliferating cells are abundant on the surface, in the cortex, and also in the medulla.

Early studies assumed that the ovary develops from the cortex of the gonad, and the testis—from the medulla. However, this is oversimplified. The medulla continues to develop and grow in both sexes. In the testes, medullary cords enlarge, forming testis cords that give rise to seminiferous tubules. The number of germ cells increases within these cords, while it diminishes in the medullary cords in the ovaries. Therefore, it is likely that genes involved in sex determination also control germ cell count in the medulla. Considering the fate of the medulla in the developing ovaries, the leopard gecko in our study exhibited well-developed medullary cords in the ovaries, where lumen formation was seen. Germ cells were no longer present within these cords. In the mourning gecko, the medullary cords were present in the developing ovaries, but they were less distinct compared with those in the leopard gecko. However, lumen formation and the presence of germ cells were noted within these cords. In the crested gecko, neither cords nor germ cells were present in the medulla of the developing ovaries. This suggests that different species have different tendencies in terms of medullary cord formation and the elimination of germ cells within the ovarian medulla. In many reptile species, the lumen within the medullary cords significantly enlarges at later stages of ovarian development, creating lacunae, that is, free spaces within the ovary. This phenomenon was described in amphibians, lizards, snakes, turtles, crocodiles, and birds [11,27,36,37,38,39]. The formation of such spaces within the ovaries may facilitate the storage of large ovarian follicles that bulge to the ovarian cavity as they grow.

This study demonstrated that the leopard gecko exhibits very early formation of cords with lumina inside. Such an early lumen formation within the testis cords was also observed in other lizards [14,15,16,26], but not in turtles or crocodiles [6,12,34,35].

In this study, we have demonstrated how immunostaining for E-cadherin and laminin reveals important aspects of gonadogenesis that are not evident through conventional histological staining. The gonadal cortex and the medullary cords are E-cadherin-positive. The cortex is separated from these cords by E-cadherin-negative stromal cells, which form a layer rich in laminin and, consequently, in the extracellular matrix. This effectively illustrates the internal structure of developing gonads and allows for the analysis of the development of the cortex, medulla, cords, and connective tissue. Additionally, immunostaining has enabled us to observe the formation of E-cadherin-positive efferent ducts, which develop in both sexes. It is not clear whether efferent ducts form by elongation of the medullary cords from the gonad to the mesonephros or if they have another origin, such as from mesonephric mesenchyme.

One of the unique features of gonadal development found only in the geckos studied here is the direction of gonadal growth. Since the onset of sex differentiation, the gonads grew medially and dorsally, and this was the direction in which their mesogonia curved (Figure 12H).

Another characteristic feature of gonadal development in geckos is the specific shape of the ovarian cortex. In the developing ovaries, the cortex was thickened on the ventral and medial surfaces of the gonad, while it remained thin on the dorsal side. This shape of the cortex was not previously described in other reptiles, suggesting that this may be specific to the Gekkota group.

The shape of the developing testes in the leopard gecko is another unique characteristic (Figure 12F). The testes were flattened bilaterally, which was not previously described in other reptiles.

Previous studies have focused on the region located between developing gonads [40,41]. We previously described the presence of a thickened coelomic epithelium extending from the gonads to the dorsal mesentery [40]. In *L. olivacea*, the expression of the *Sox9* gene in this region was reported, referred to as the T-domain (T-dom) [41]. In the geckos studied here, we did not find a thickening of the coelomic epithelium in this region, and future investigations should involve immunolocalization of SOX9 in this region to verify whether the T-domain is present in reptiles other than *L. olivacea*.

Among lizards, the genes and mechanisms determining sex have been poorly studied so far. This refers both to species with genetic sex determination and those with temperature sex determination. Among lizards with genetic sex determination, the crested gecko may serve as a potential model species due to the ease of breeding and reproduction. Among lizards with temperature sex determination, the leopard gecko might be a good candidate. As mentioned above, temperature affects sex determination during the thermosensitive period. In the case of the sea turtle (*L. olivacea*), this occurs during the undifferentiated gonadal stage [6]. In the leopard gecko, the thermosensitive period also most likely occurs during this phase of gonadogenesis, which corresponds to the period between S29 and S32. Experiments involving temperature or hormonal manipulations should be conducted during these developmental stages. Finally, the mourning gecko, despite being a parthenogenetic species, might be used in experiments investigating whether temperature or hormonal factors can induce testicular development in this species.

## 5. Conclusions

This study identified a series of unique features in the gonadal development specific to the examined species of geckos, as well as a series of common features in the development of their gonads and other vertebrates. The earliest sign of the gonadal formation in three studied gecko species was the appearance of two bulges of the thickened coelomic epithelium located at both sides of the dorsal mesentery as observed in all vertebrates. These ridges formed on the dorsal end of the dorsal mesentery, rather than on the surface of the mesonephros. The earliest gonadal ridges contained a rudimentary cortex continuous with a coelomic epithelium, similar to all vertebrates. Mesenchyme accumulated in the center of the ridges. In the gonadal medulla, medullary cords formed, originating as protrusions from the cortex. As in other vertebrates, the sexual differentiation of gonads in the geckos studied here was evident in the thickening of the ovarian cortex and an increase in the number of germ cells. In differentiating testes, the cortex became thin and lacked germ cells. A distinctive feature of the studied species was the rapid development of testis cords and the early appearance of lumens within them. The medulla of the ovaries exhibited varied tendencies to organize cells into medullary cords, with well differentiated cords in the leopard geckos. A characteristic feature of the developing gecko ovaries was the thickened crescent-shaped cortex on the medial and ventral surfaces. Another notable feature of embryonic gecko gonads was their growth in a horizontal plane.

## Figures and Tables

**Figure 1 biology-13-00007-f001:**
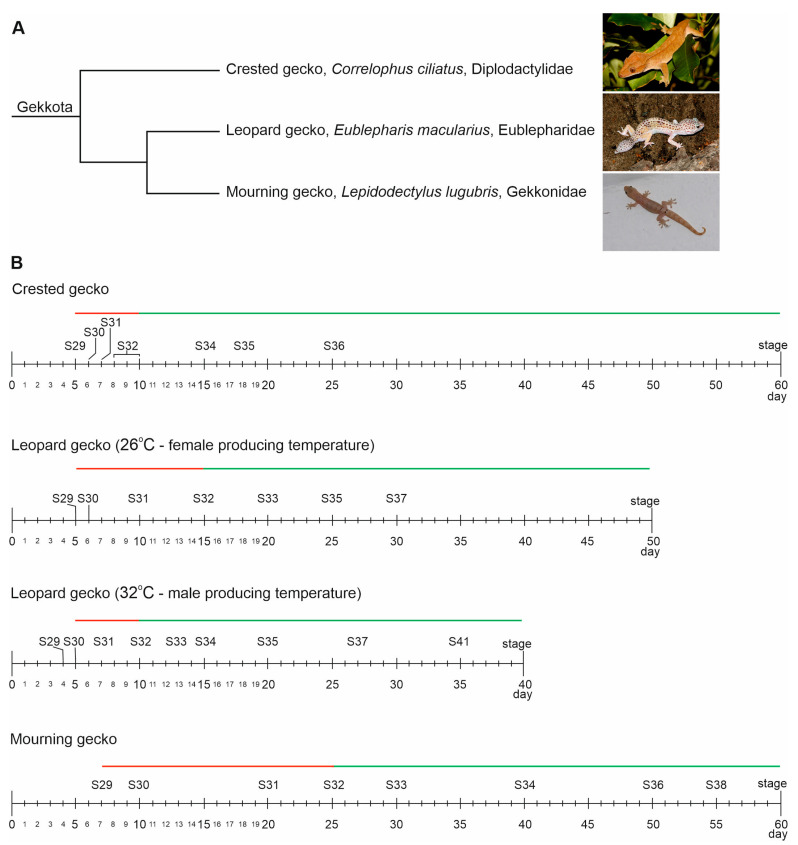
(**A**). Phylogenetic relationships between the studied gecko species. (**B**). The timeline and developmental stages investigated in the study. The red line indicates the phase of undifferentiated gonads, while the green line represents the phase of differentiated gonads. The sources of images: https://commons.wikimedia.org/wiki/File:Crested_gecko_-_1.jpg, CC-BY-4.0; https://commons.wikimedia.org/wiki/File:Eublepharis_macularius_fg01.JPG, CC-BY-SA-2.5; https://commons.wikimedia.org/wiki/File:Lepidodactylus_lugubris_1.JPG, CC-BY-SA-3.0.

**Figure 2 biology-13-00007-f002:**
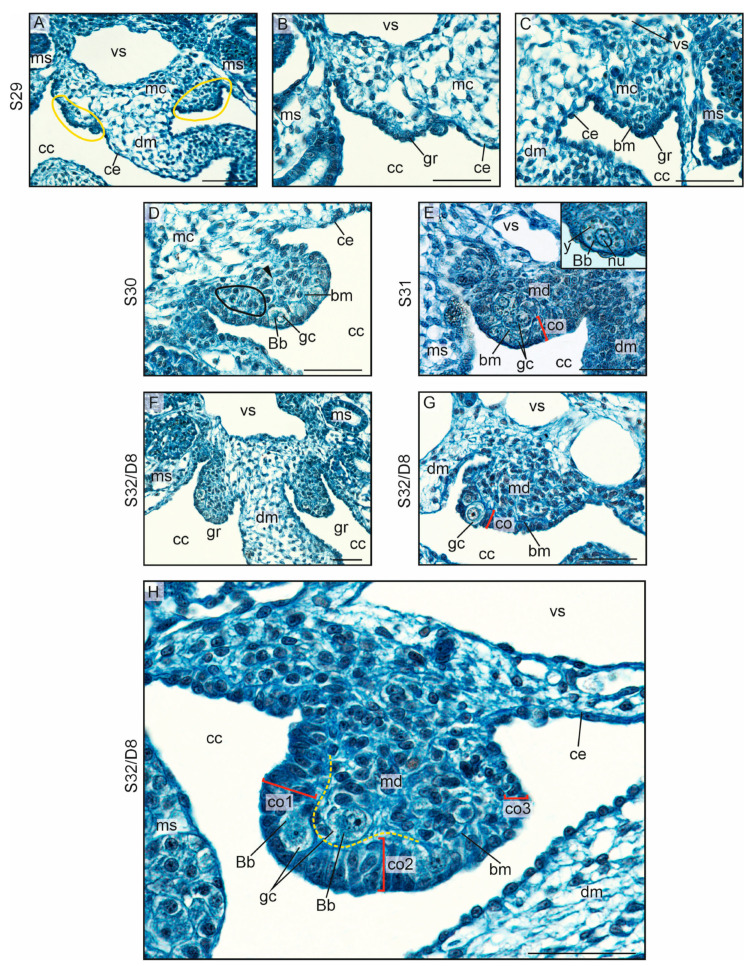
Gonadal ridges and undifferentiated gonads of the crested gecko (*Correlophus ciliates*). (**A**–**C**). The gonadal ridges (encircled) at stage 29 (S29) and embryonic day 5 (D5) are visible as two bulges protruding to the coelomic cavity (cc) at both sides of the dorsal mesentery (dm). The thickened epithelium consists of the surface of the gonadal ridges (gr), lies on the basement membrane (bm), and is continuous with the coelomic epithelium (ce). The space above the ridges is filled with loose mesenchymal tissue (mc). The mesenchymal cells (mc) aggregate inside the gonadal ridges (**C**). Mesonephroi (ms) and blood vessel (vs) are located dorsally to the gonadal ridges. (**D**). At S30/D6, the epithelium of the gonadal ridge is thicker than at S29. The first germ cells (gc) are present in the superficial epithelium of the gonadal ridge. Germ cells can be easily recognized due to their large size, spherical cell shape, and prominent nucleus. In the cytoplasm, a Balbiani body (Bb) is visible as a blue cloud, and at such an early stage, the yolk is present in the form of small yellow droplets. On the opposite side of the basement membrane (bm), groups of somatic cells arranged in the form of cords (encircled) are visible, surrounded by their own basal membranes (arrowhead). The beginning of the differentiation into the primitive cortex (surface epithelium of the gonad) and medulla (cords in the center of the gonad) can be seen. (**E**). At S31/D6, the germ cells (gc) are more numerous and are present both in the cortex (co) and in the medulla (md). Insert: the germ cells with centrally located nucleus (nu), yolk droplets (y), and Balbiani body (Bb). (**F**–**H**). At S32/D8, the gonads are larger and protrude more prominently into the coelomic cavity. The medulla is the part of the gonad that is noticeably enlarged. (**H**). A larger magnification showing the details of the undifferentiated gonadal structure. The division into cortex (co), which is the surface epithelium, and medulla (md) located in the center of the gonad, is visible. The basement membrane (bm) is a border between the cortex and medulla, indicated by the yellow dashed line. Different thickness of the cortex can be seen in the cross-section. The cortex is thickest on the lateral side (co1) (facing the mesonephros, ms) and the ventral surface (co2). The epithelial cells are elongated; the epithelium appears columnar in these regions, and there are germ cells (gc) present. On the other hand, on the medial side, facing the dorsal mesentery (dm), the cortex (co3) is devoid of germ cells. It is thin and takes the form of cuboidal epithelium. Scale bar, 50 μm.

**Figure 3 biology-13-00007-f003:**
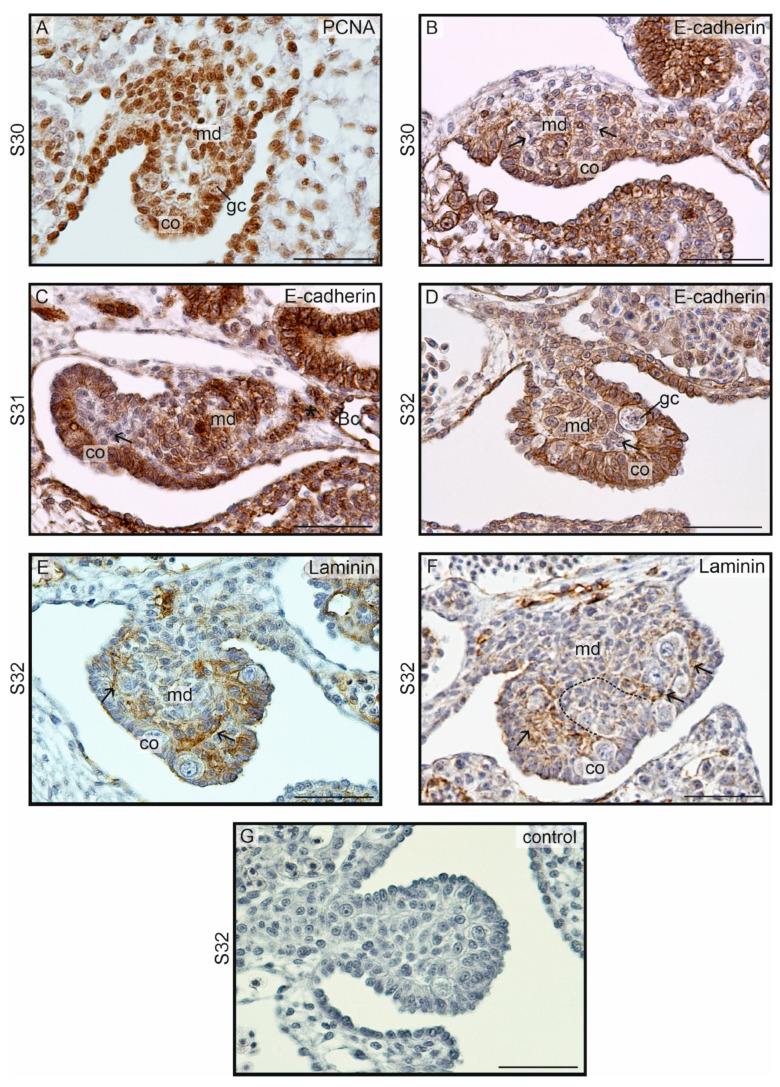
Immunostaining in the undifferentiated gonads in the crested gecko (*Correlophus ciliatus*). (**A**). Immunostaining of PCNA at S30/D6. Germ (gc) and somatic cells in both cortex (co) and medulla (md) show strong PCNA-positive signal. (**B**). Immunostaining of E-cadherin at S30; E-cadherin positive or negative cells correspond to epithelial or stromal gonadal tissues, respectively. From the beginning of the medullary cord (md) formation, its cells are E-cadherin-positive (E-cadherin+), similarly to the primordial cortex (co). In the center of the gonad also E-cadherin-negative (E-cadherin–) cells are present (arrow). (**C**,**D**). At next stages of undifferentiated gonads E-cadherin signal is visible in the cortex and medullary cords. Both are separated by E-cadherin– stromal cells (arrows). A primordium of efferent ducts (asterisk) are also E-cadherin+ and forms a stream of somatic cells reaching the nearest Bowman’s capsule (Bc). (**E**,**F**). Immunostaining of laminin at S32. The space filled with stromal cells (arrows) shows the signal of laminin. The dashed line (**F**) indicates the outline of the developing medullary cord; it is not separated by a layer containing laminin from the cortex. (**G**). Negative control. Scale bar, 50 μm.

**Figure 4 biology-13-00007-f004:**
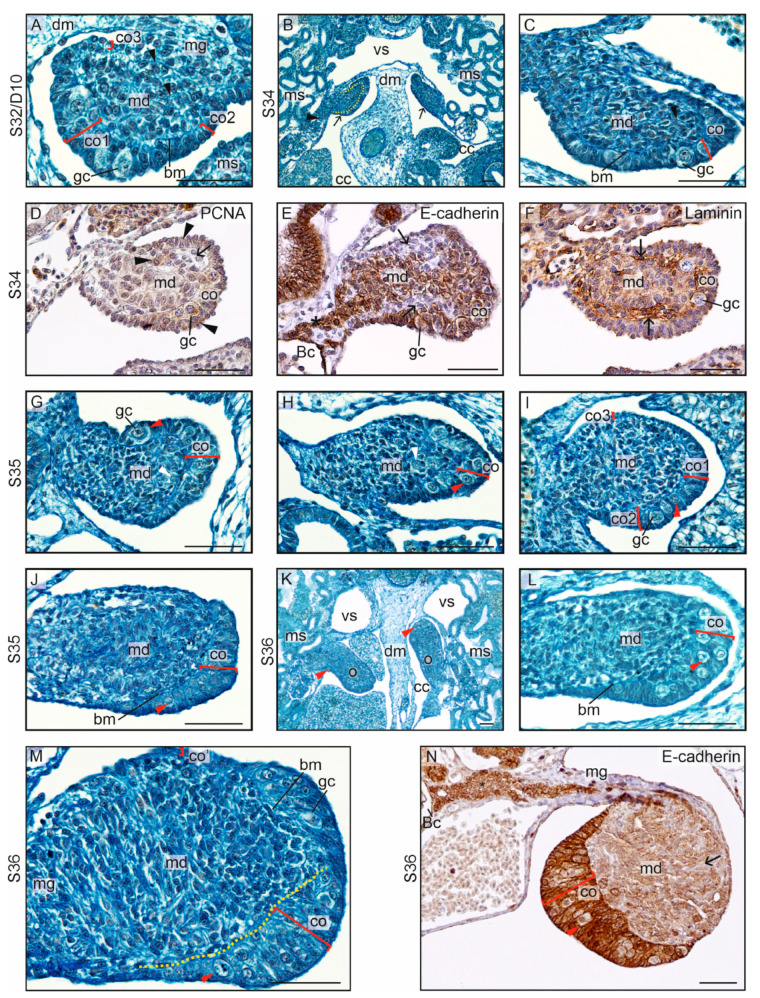
Ovarian differentiation in the crested gecko (*Correlophus ciliates*). (**A**). In the differentiating ovaries, at stage 32 (S32) and day 10 (D10), the division into the thick cortex (co) containing germ cells (gc) and the centrally located medulla (md) is evident. The basement membrane (bm) separates the cortex and medulla. Smaller basement membranes (arrowheads) enclose the cord-like groups of cells in the medulla, the medullary cords. The cortex (co1) is thickest at the distal pole of the gonad located on the opposite side of the mesogonium (mg). The mesogonium on which the gonad is suspended becomes thinner. The cortex (co2) of the ventral region, facing the mesonephros (ms), is thinner than co1. However, the cortex (co3) of the dorsal region, facing the dorsal mesentery (dm), is the thinnest. The gonad begins to bend towards the dorsal mesentery and in a medial-dorsal direction. (**B**). At S34/D15, the tendency of gonadal growth (indicated by arrows) in the medio-dorsal direction is evident. The yellow dashed line indicates the border between the cortex and medulla. The medulla consists of a large cluster of somatic cells that extends up to the adjacent nephric bodies. The arrowhead indicates the location where the cluster can be found close to the nephric body. The coelomic cavity (cc) and the blood vessel (vs) are indicated. (**C**). A larger magnification of the differentiating ovary at S34/D15. The main basement membrane (bm) separates the cortex (co) and medulla (md). Smaller basement membranes (arrowhead) are also present in the medulla. Germ cells (gc) are present only in the cortex. (**D**). Immunostaining of PCNA in differentiating ovary at S34. PCNA+ cells (arrowheads) are present in the cortex and medulla, but not in the stroma (arrow). (**E**). Immunostaining of E-cadherin. Strong E-cadherin staining is visible in the cortex (co), medulla (md) and in the streak of the somatic cells (asterisk) reaching from the gonad to the Bowman’s capsule (Bc) in the mesonephros. Stromal cells (arrows) filling the space between the cortex and medulla are negative for E-cadherin. (**F**). Immunostaining of laminin reveals the strongest accumulation of this protein in the stroma (arrows) between the cortex and medulla. (**G**–**I**). At S35/D17, the division into the cortex and medulla is evident. There are numerous germ cells (gc) in the cortex. Only sporadically, individual germ cells (white arrowheads) are found in the medulla. (**J**). The developing ovary at S35/D19; the basement membrane (bm) separates the cortex (co) and medulla (md). (**K**,**L**). The ovaries (o) at S36/D22, and magnification of one ovary at the same stage. (**M**). The ovary at S36/D25. The characteristic crescent shape of the ovarian cortex is visible, with the thickest cortex (co) at the distal pole of the gonad located on the opposite side of the mesogonium (mg) and gradually thinning (co’) towards the mesogonium. (**N**). Immunostaining of E-cadherin in the ovary at S36. The strongly stained E-cadherin+ cortex (co) contains bright germ cells. In the extensive medullary regions (md), streaks of brown medullary cells interspersed with streaks of light stromal cells (arrow) are present. Within the mesogonium (mg), a streak of strongly stained E-cadherin+ cells is present (asterisk), extending to the Bowman’s capsule (Bc). This represents the primordial efferent ducts. Scale bar, 50 μm.

**Figure 5 biology-13-00007-f005:**
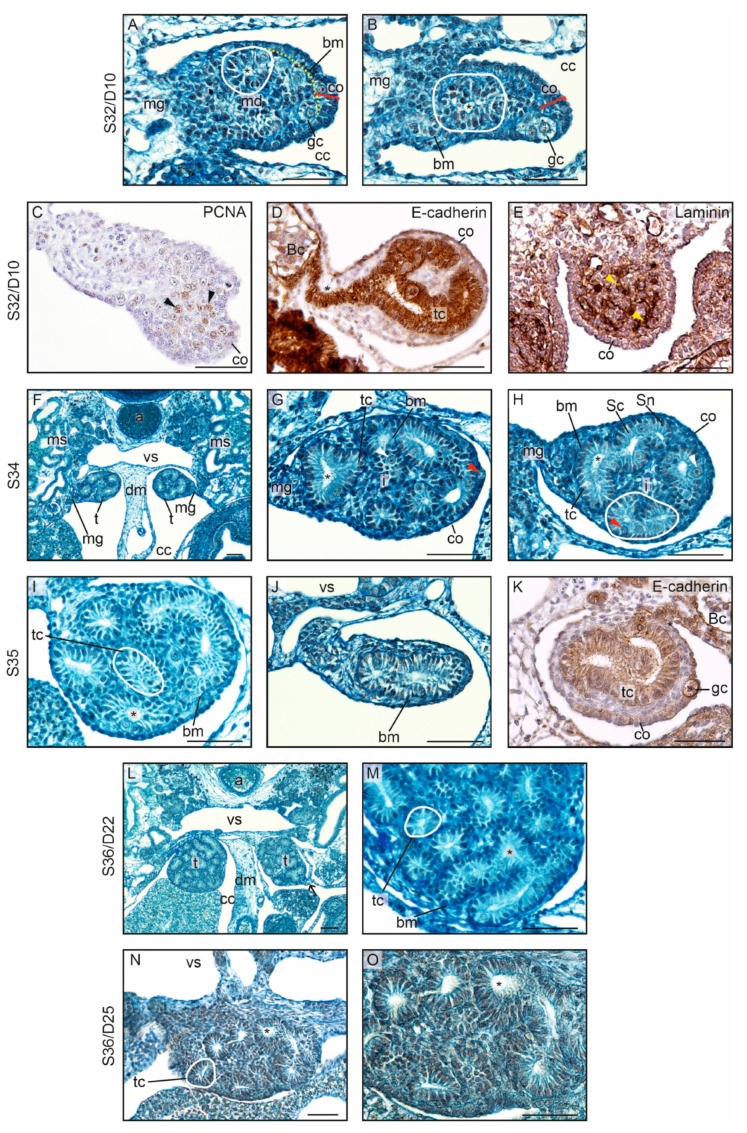
Testicular differentiation in the crested gecko (*Correlophus ciliates*). (**A**,**B**). The differentiating testes at stage 32 (S32) and day 10 (D10) are characterized by a thin cortex (co) with only few germ cells (gc) and a more distinct formation of cords in the medulla (md) compared with the differentiating ovaries. The yellow dashed line indicates the border between the thinning cortex and medulla. In (**B**), the basal membrane (bm) is evident, separating the cortical part from the medullary part. The testis cords within the medulla have a wall composed of a differentiating epithelium, and a lumen forms in the center of the cords (indicated by asterisks). The testis is located in the coelomic cavity (cc) and suspended on the mesogonium (mg) (black arrow). (**C**) Immunostaining of PCNA in differentiating testis at S32. PCNA+ cells (black arrowheads) are dispersed in the interior of the gonad. (**D**). Immunostaining of E-cadherin in differentiating testis at S32. Strongly E-cadherin-positive testis cords (tc) are visible in the gonad. The cords are continuous with the streak of E-cadherin+ cells (asterisk) in the mesogonium. The streak reaches the Bowman’s capsule (Bc) and constitutes a primordium of the efferent ducts. Stromal cells, negative for E-cadherin, fills the space between the thin cortex (co) and the testis cords. (**E**). Immunostaining of laminin. Streaks of laminin (yellow arrowheads) are dispersed in the medulla, marking the stroma. (**F**). The mediodorsal orientation of the growing testes (t) at S34/D15; a, aorta; ms, mesonephros; dm, dorsal mesentery; vs, blood vessel (subcardinal vein). (**G**,**H**). At S34/D15, the testis cords (tc; encircled) are distinct in the testes. The wall of the testis cords is composed of a row of orderly arranged epithelial cells (Sertoli cells, Sc, with basally located nuclei, Sn), with germ cells (white arrowheads) located between them. At this stage, a distinct lumen is present in each cord. Interstitial tissue (i) forms between the cords. Occasionally, singular germ cells (red arrowhead) are found in the cortex (co), which at this stage have already transformed into a thin simple epithelium. (**I**–**K**). Testes at S35/D17. (**I**). A cross-section through the middle segment of the gonad. Within the testis cords, a lumen is present (asterisk). (**J**). A cross-section through the posterior end of the gonad. The gonad is small and only one testis cord is visible. (**K**). Immunostaining of E-cadherin in the testis. E-cadherin+ testis cords (tc), cortex (co) and primordium of the efferent ducts (asterisk) are visible. (**L**–**O**). At S36/D22 and S36/D25, an increase in the number and length of testis cords can be observed. The arrow (**L**) indicates the mesogonium, and the gonad is oriented in a mediodorsal direction. Scale bar, 50 μm.

**Figure 6 biology-13-00007-f006:**
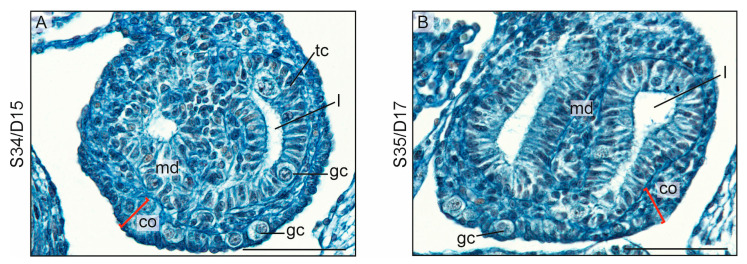
Intersex gonads of two crested geckos (*Correlophus ciliates*). (**A**,**B**). Intersex gonads are characterized by a smaller number of testis cords (tc) in the medulla (md). However, these cords still have a lumen (l), as in typical testes. Another characteristic feature is the presence of a cortex (co) that is thicker than in typical testes. Within this cortex, relatively numerous germ cells (gc) can be found. Scale bar, 50 μm.

**Figure 7 biology-13-00007-f007:**
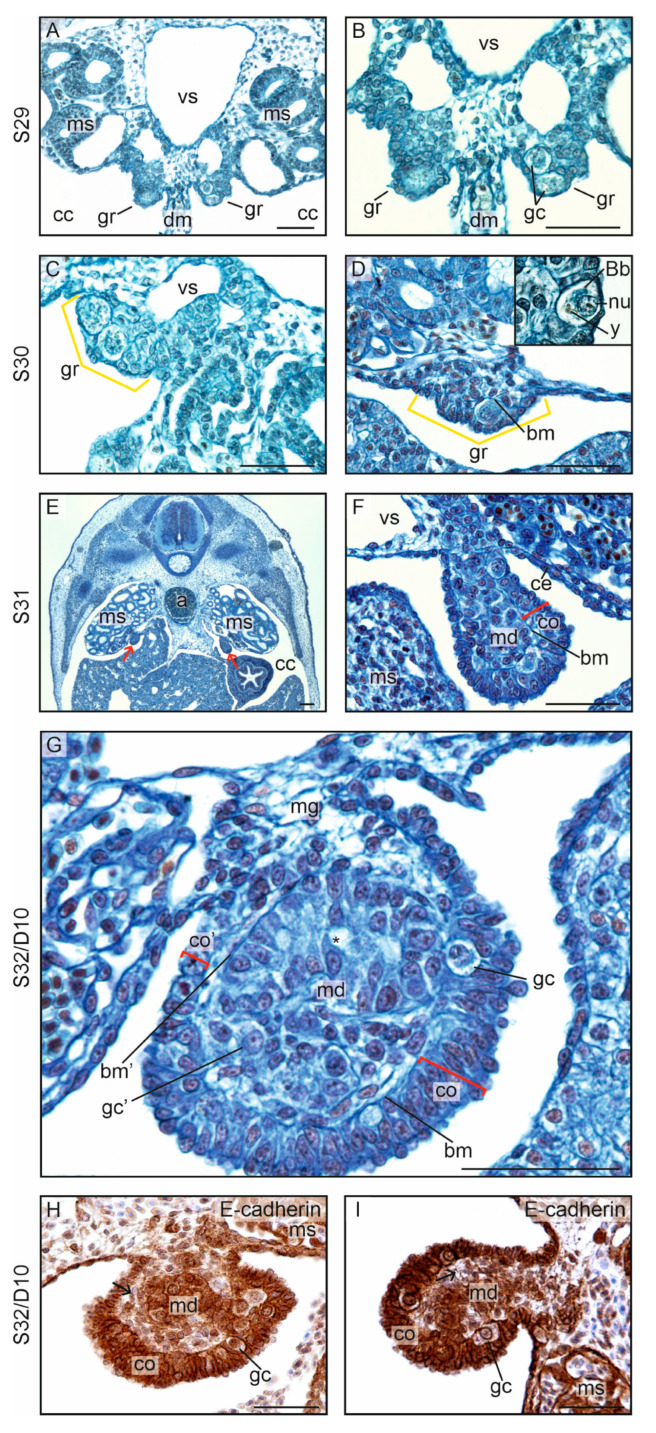
Gonadal ridges and undifferentiated gonads of the leopard gecko (*Eublepharis macularius*). (**A**,**B**). At stage 29 (S29) and embryonic day 11 (D11), the gonadal ridges (gr) are visible as two bulges protruding into the coelomic cavity (cc). Typically, the dorsal mesentery (dm) is located between the gonadal ridges, and the mesonephroi (ms) and blood vessels (vs) are positioned above them. The first germ cells (gc) are present in the gonadal ridges. (**C**,**D**). At stage S30/D11, the gonadal ridges (marked with yellow brackets) continue to take the form of gonadal ridges, which are in close contact with the mesonephros along a broad surface. The somatic cells are more numerous compared with S29, and the basement membrane (bm) indicates a division into the primitive cortex and medulla. Insert: the germ cells with centrally located nucleus (nu), yolk droplets (y), and Balbiani body (Bb). (**E**,**F**). At S31, the position of undifferentiated gonads (arrows) on the ventromedial surface of the mesonephros can be seen in (**E**); a, aorta. A closer magnification (**F**) reveals that the gonad is suspended on the mesogonium (mg), and there is a clear division into the cortex (co) and medulla (md). These regions are separated by the basement membrane (bm). ce, coelomic epithelium. (**G**). At S32/D10, the thickest cortex (co) is present at the distal pole of the gonad located on the opposite side of the mesogonium (mg). The closer to the mesogonium, the thinner the cortex (compare co and co’). Germ cells are present both in the cortex and in the medulla (gc and gc’, respectively). In addition to the basement membrane (bm) separating the cortex from the medulla, there are also basement membranes (bm’) within the medulla, surrounding the clearly developing cords. The small spaces (asterisk) visible in the center of the cords indicate the initiation of lumen formation inside the cords. (**H**,**I**). Immunostaining of E-cadherin in the undifferentiated gonads at S32/D10. A strong signal of E-cadherin is visible in the cortex (co) and in the medullary cord (md). The medullary cord joins the cortex. E-cadherin– stroma (arrow) fills the space between the cortex and the cord. Scale bar, 50 μm.

**Figure 8 biology-13-00007-f008:**
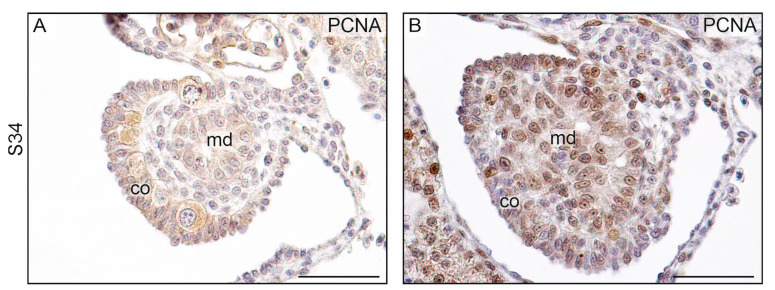
Immunostaining of PCNA in the differentiating gonads in the leopard gecko (*Eublepharis macularius*). (**A**). The thick cortex with clearly visible germ cells here indicates ovarian differentiation at S34. Signal of PCNA is visible in the cortex (co). (**B**). Differentiating testis with a sterile cortex (co) at S34 with visible signal in the cortex and medulla (md). Scale bar, 50 μm.

**Figure 9 biology-13-00007-f009:**
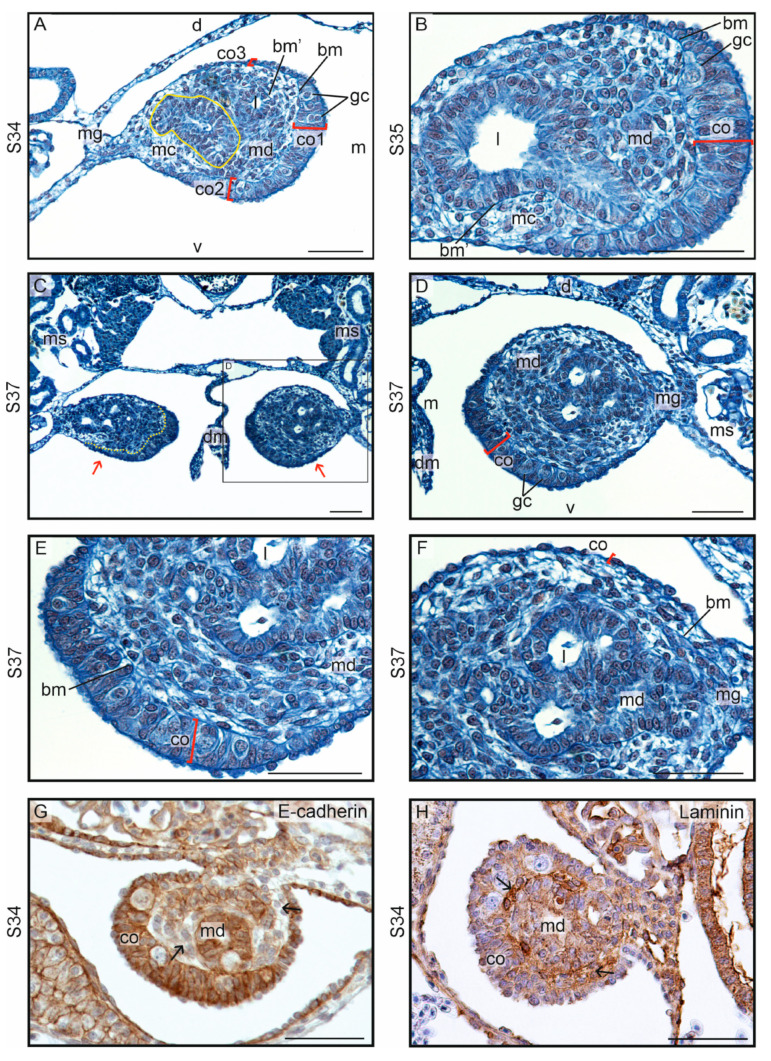
Ovarian differentiation in the leopard gecko (*Eublepharis macularius*). (**A**). At stage 34 (S34), the differentiating ovary is characterized by a thick cortex (co) containing germ cells (gc). The cortex (co1) is the thickest at the end of the gonad, farthest from the mesogonium (mg) and on the ventral (v) surface of the gonad (co2). However, it is thinnest on the dorsal (d) surface (co3). The distal tip of the gonad is oriented towards the medial direction (m). A distinct basement membrane (bm) marks the boundary of the cortex. Within the medulla, there are no germ cells; however, there are clear cords (encircled) surrounded by the basement membrane (bm’), and a lumen (l) begins to develop within them. In the cross-section, one or two cords can be observed in the medulla, along with a significant amount of dispersed mesenchyme (mc). (**B**). The ovary at S35 with one large cord in the medulla (md) and a voluminous lumen (l) inside the cord. The germ cells (gc) are located only in the cortex (co). (**C**). A horizontal orientation of the ovaries (indicated by red arrows) at S37/D30. The yellow dashed line indicates the border between the cortex and medulla. Mesonephroi (ms) are located dorsally relative to the gonads, and the dorsal mesentery (dm) is located between the gonads. (**D**–**F**). Detailed structures of the ovaries at S37/D30. The characteristic crescent-shaped cortex is visible. A thick cortex is observed in the medial and ventral regions (**E**), and a thin cortex is visible on the dorsal surface of the gonad (**F**), transformed into a simple squamous epithelium. The medulla contains cords with well-developed epithelium and a lumen (l) in the center. (**G**). Immunostaining of E-cadherin in the ovary at S34. A layer of E-cadherin– stromal cells (black arrows) is present between the E-cadherin+ cortex (co) and E-cadherin+ medullary cords (md). (**H**). Immunostaining of laminin in the ovary at S34. A layer of E-cadherin– stromal cells (black arrows) shows a presence of laminin. Scale bar, 50 μm.

**Figure 10 biology-13-00007-f010:**
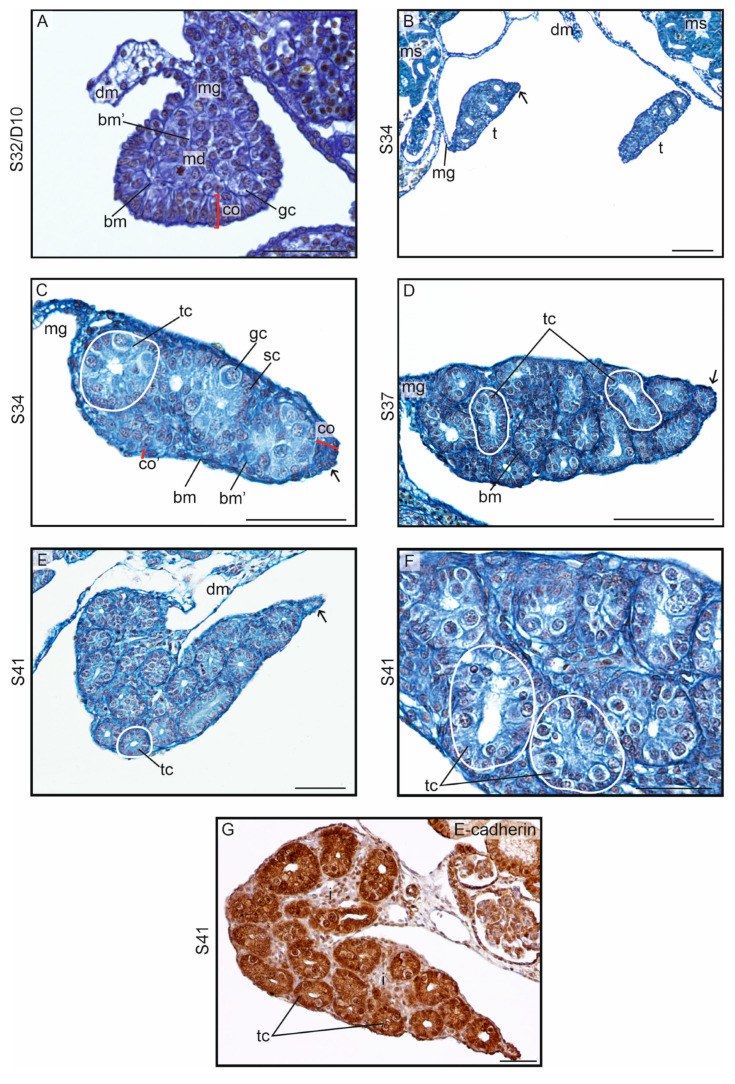
Testicular differentiation in the leopard gecko (*Eublepharis macularius*). (**A**). At stage 32 (S32) and day 10 (D10) after laying, the gonad is composed of the cortex (co) and medulla (md) separated by a basement membrane (bm). The presence of germ cells in the medulla suggests the initiation of testicular differentiation. Within the medulla, basement membranes (bm’) are formed, indicating the beginning of cord formation; dm, dorsal mesentery. (**B**). At S34, the testes (t) tend to grow vertically and are suspended on narrow mesogonia (mg). The testes are clearly bilaterally flattened. The spaces within the testes are clearly visible, representing the lumen in the center of the testis cords. The arrow indicates the distal tip of the gonad. ms, mesonephros. (**C**). The magnification of the testicular structure at S34. The encircled structure is a testis cord (tc) with a lumen in the center; the cord wall is formed by a row of epithelial cells (differentiating Sertoli cells, sc), with germ cells (gc) embedded in between. The cortex is transformed into a simple squamous epithelium (co’), which remains slightly thicker at the distal tip (co). (**D**–**F**). In the subsequent stages (S37 and S41), testicular growth is noted, and an increasing number of testis cords are visible. At S37 (**D**), a well-differentiated epithelium within the testis cords is apparent, and the cords are surrounded by clearly visible, complete basement membranes. The testes at these stages remain bilaterally flattened, with a tapering at the distal end. (**G**). Immunostaining of E-cadherin in the testis at S41. E-cadherin+ testis cords (tc) are separated by E-cadherin– interstitium (i). Scale bar, 50 μm.

**Figure 11 biology-13-00007-f011:**
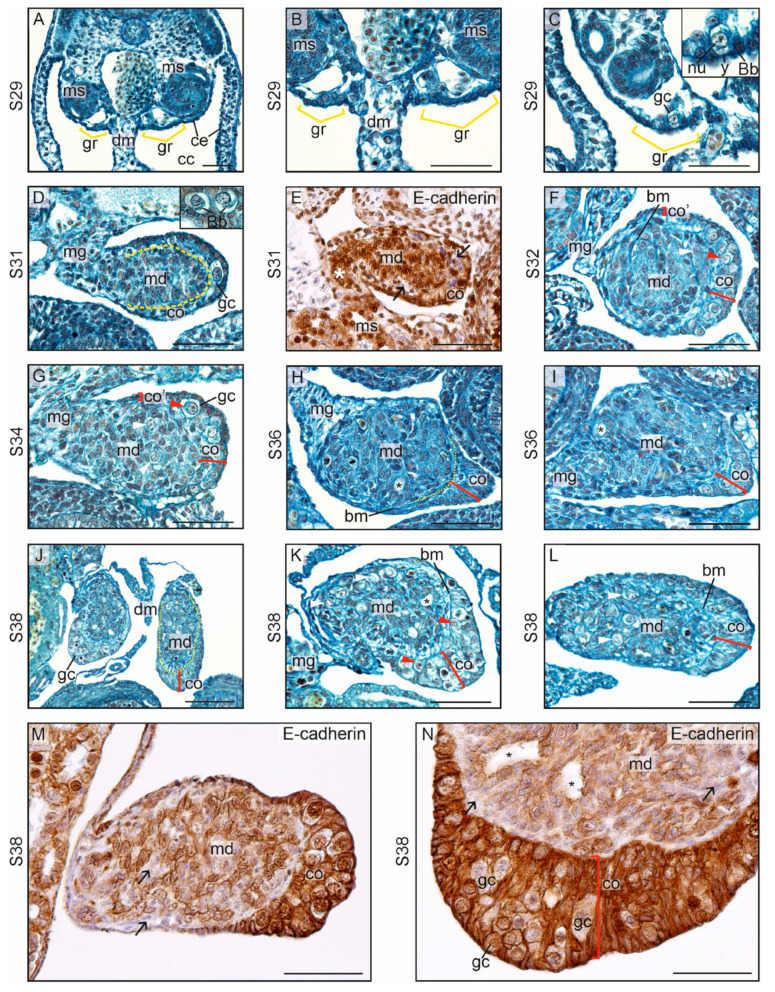
Gonadal development in the mourning gecko (*Lepidodactylus lugubris*). (**A**–**C**). At stage 29 and day 7 (S29/D7), the thickening of the coelomic epithelium (ce) in the ventral position relative to the mesonephroi (ms), on both sides of the dorsal mesentery (dm), indicates the initiation of the gonadal ridge formation (marked with yellow brackets). The first germ cells (gc) are present in the gonadal ridges (gr); the insert provides details, including the round and pale nucleus (nu), the presence of the Balbiani body (Bb), and yolk droplets (y). cc, coelomic cavity. (**D**). At stage S31/D10, a division into the cortex (co) and medulla (md) is evident. The yellow dashed line indicates the border between these two regions. First germ cells appear in the cortex. (**E**). Immunostaining of E-cadherin in the gonad at S31/D10. A layer of E-cadherin– stromal cells (black arrow) is located between the E-cadherin+ cortex (co) and medulla (md). In the mesogonium, a primordium of the efferent ducts (white asterisk) reaching to the mesonephros is visible. (**F**,**G**). During S32/D25, and S34/D40, the growth of the gonads is noted. They are suspended on a narrow mesogonium (mg) and are directed medially. Germ cells are numerous in the cortex, and an increase in cortical thickness is noted (**F**). This indicates ovarian differentiation. The crescent-shaped cortex (co) is the thickest at the distal pole of the gonad located on the opposite side of the mesogonium (mg). The closer to the mesogonium, the thinner the cortex (co’). Germ cells are present also in the medulla (red arrowhead indicates cortical germ cells, and white arrowhead—medullary germ cells). The basement membranes (bm) are present also inside the medulla and enclose the forming cords. Along with an increase in germ cell count and cortical thickness, the thickness of the ovarian cortex becomes diversified. (**H**,**I**). At S36/D50, the cortex (co) has a characteristic clustering at the tip of the gonad. The yellow dashed line points to the border between the cortex and medulla. The cords within the medulla are not clearly differentiated, but at this stage, the first spaces appear (black asterisks), indicating the initiation of lumen formation within the cords. The medulla is the dominant part of the gonad. (**J**–**L**). At S38/D55, the cortex is a voluminous cluster of cells located at the tip of the gonad, and numerous germ cells (red arrowheads) are present in this part of the gonad. Germ cells (white arrowheads) are also present in a massive medulla. (**M**,**N**). Immunostaining of E-cadherin in the ovary at S38/D55. A thick E-cadherin+ cortex (co) is visible. Germ cells (gc) are located in the cortex. In the medulla (md), a weak E-cadherin signal is visible in clusters of somatic cells separated by E-cadherin-negative stromal cells (black arrows), indicating the presence of loosely organized cords in the ovarian medulla; within E-cadherin-positive clusters, lumens are present (asterisks). Scale bar, 50 μm.

**Figure 12 biology-13-00007-f012:**
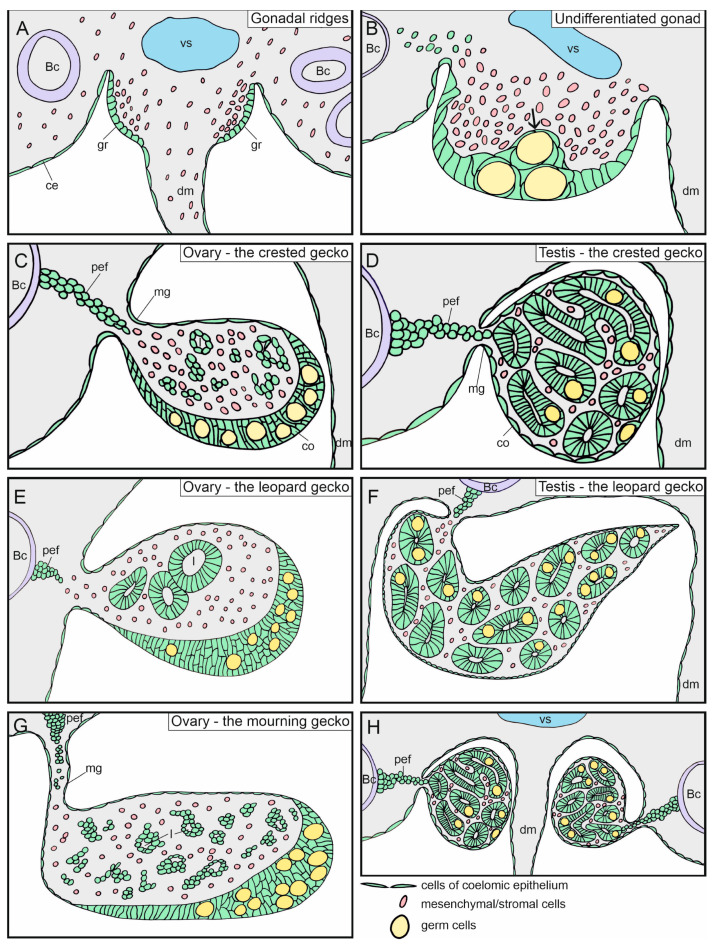
Schematic comparison of developing gonads in geckos. (**A**). The gonadal ridges (gr), as a pair of thickened coelomic epithelium (ce, green cells) located of three studied gecko species on the dorsal end of the dorsal mesentery (dm). The space located inwards from the coelomic epithelium is filled with accumulated mesenchymal cells (pink). (**B**). In the undifferentiated gonad of geckos, a division into cortical and medullary regions is visible; the cortical region is formed by a thickened epithelium continuous with the coelomic epithelium, while the medullary region is centrally located; germ cells (yellow) are present within the cortical epithelium and in developing medullary cords (arrow) that protrude from the cortex; blood vessel (vs) and Bowman’s capsules (Bc) are in the vicinity of the gonad. (**C**). In the differentiating ovary of the crested gecko, a thick cortex (co) is evident in the ventral and medial regions, while it is thin at the gonadal hilus (mesogonium, mg) and on the dorsal side of the gonad; underdeveloped structures resembling medullary cords with lumen (l) inside are present in the medulla; germ cells are located within the cortex; cells forming the primordium of efferent ducts (pef) extend from Bowman’s capsules to the mesogonium. (**D**). In the differentiating testis of the crested gecko, the cortex (co) is transformed into a thin epithelium covering the gonad; in the medullary region, numerous testis cords are present, exhibiting lumen and germ cells inside; the primordium of efferent ducts (pef) is present. (**E**). In the differentiating ovaries of the leopard gecko, a similar degree of gonadal cortex development is observed as in the crested gecko; however, advanced medullary cords with luminal spaces are present in the medullary region. (**F**). The differentiating testes of the leopard gecko also exhibit a similar degree of internal structure development as observed in the crested gecko; however, their shape is bilaterally flattened, and they have a pointed shape on the cross section. (**G**). In the differentiating ovaries of the mourning gecko, the gonadal structure is similar to that of the crested gecko, with underdeveloped medullary cords. (**H**). The illustration depicts growth of developing gecko gonads in a horizontal plane.

## Data Availability

Data are contained within the article and Appendix A.

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
