# Peer review of "Histological Analysis of Gonadal Ridge Development and Sex Differentiation of Gonads in Three Gecko Species"

_biology, 2023, doi:10.3390/biology13010007_

Round 1
Reviewer 1 Report
Comments and Suggestions for Authors
Comments to the Introduction
Lines 51-56
Authors: “However, this is an oversimplification. A large portion of the ovary called the stroma, also develops from the medulla, while the cortex in the testis forms the epithelium covering the male gonad. Within the ovarian cortex, germline nests develop, which then fragment into ovarian follicles. In the testicular medulla, on the other hand, testis cords develop and transform into seminiferous tubules.
My comment:
Witchi’s classical studies in amphibia did not consider the stromal tissue in the ovary. However, in developing gonads of Amniota vertebrates, stromal tissues are mostly formed by mesenchymal cells, fibroblasts, and blood vessels.
Comments to Material and Methods
Figure 2
My comment:
The Balbiani body (Bb) of germ cells is shown only in Fig. 2G. It would be useful to show it in Fig. 2H also. I suggest the authors mention the organelle composition of the (Bb).
Figure 3
Figs. 3C and 3D
The authors fail to mention that E-cadherin positive or negative cells correspond to epithelial or stromal gonadal tissues. Laminin is indicated only in the cortex of Fig. 3E (arrows); however, the thinner positive mark around the medullary cords is not mentioned. This point is important as the epithelial origin of the medullary cords from the surface epithelium was found in the genital ridges of turtles and mammals.
Figure 4
Fig. 4D
Proliferating cells are not indicated.
Figs. 4E and 4F
My comments:
Although E-cadherin-positive epithelial tissues are evident in 4E, the surrounding laminin-positive medullary cords in 4F are not correctly mentioned. The authors mention that “laminin strongly accumulates in the stroma.” This point is important because the basal lamina, which segregates epithelial from stromal tissues, is important in establishing the undifferentiated (bipotential) gonads before their dimorphic sexual morphogenesis as testes or ovaries.
Figure 5
Fig. 5E lines 369-361.
Authors: “Laminin immunostaining showed that gonadal stroma was dispersed in differentiating testes and did not form one layer between the cortex and medulla as in the ovaries.”
My comment: The protein laminin is produced by epithelial cells and then deposited on the surface facing the stromal tissue.
Figure 7
Lines 430-433.
Authors: “Moreover, the gonadal ridges in the leopard gecko also had a form of two bulges composed of: 1) a primitive cortex derived from a thickened coelomic epithelium and containing germ cells; and 2) a primitive medulla as an accumulation of mesenchymal cells located inside the gonadal ridge (Fig. 7D; gr, gonadal ridges).”
My comment: The “accumulation of mesenchymal cells” in the genital ridges might be incorrect because their differentiation state cannot be yet ascertained.
Lines 435-437
“The primitive medulla in both species was a cluster of somatic cells continuous with the mesonephric mesenchyme, which suggests that the gonadal medulla was formed by the aggregation of mesenchymal cells.”
My comment: Unfortunately, I think there is confusion in using the term “medulla.” As a spatial gonadal site, the gonadal medulla contains the epithelial medullary cords and their surrounding stromal tissue formed by mesenchymal cells, fibroblasts, and blood vessels. If the medullary cords are formed by “aggregation” of mesenchymal cells, one must assume that, in contrast with mammals and turtles, a mesenchymal to epithelial transition occurs in geckos.
Figure 9
Lines 512.513
“The zone of E-cadherin–stromal cells contained laminin, indicating their mesenchymal nature (Fig. 9H).”
However, the authors again assume a Mesenchymal-epithelial transition may occur in geckos. However, Figure 9G shows a clear continuity of E-cadherin-positive cells among the epithelial cortex and a medullary cord, suggesting their common origin.
Figure 10
Lines 583-584: “Yolk droplets were present when the germ cells invaded the gonadal ridges at S29, but the yolk disappeared soon after as in other geckos.”
My comment: Although this is an interesting physiological detail, no pictures are shown.
Figure 11
Figs.11M and 11N. Although the authors mention the presence of “dispersed streams of E-cadherin+ somatic cells,” in my opinion, they correspond to medullary cords.
Authors: Lines 589-581: “Immunostaining showed the presence of a cluster of E-cadherin+ cells in the medulla. This cluster extended to the mesonephros (Fig. 11E). A layer of E-cadherin–stromal cells (black arrows on Fig. 11E) was present between the medulla and the cortex, both positive for E-cadherin.
My comment: In this paragraph, the authors clearly distinguish between the two kinds of tissues in the medulla of developing gonads: (1) E-cadherin-positive medullary cords and (2) E-cadherin-negative stromal cells. I suggest replacing +/- with positive/negative.
Discussion
Lines 661-662:
Authors: “This indicates that at such an early stage, the initiation of gonadal development is synchronized with somatic development.”
My comment: This is an interesting point; however, the precise correlation is between the paired buds and the establishment of the genital ridges rather than “with somatic development.”
Lines 686-689
Authors: “In this study, we have demonstrated that in geckos, medullary cords are connected to the cortex, indicating that they form as an outgrowth of the cortex, similar to what has been shown 688 in T. scripta [10]. This indicates the origin of the medullary cords from the gonadal cortex.”
My comment: This paragraph supports my comments on the author’s figures interpretation in their Introduction and Results. The epithelial origin of medullary cords from the mesothelial cells has also been shown in other amniotes as the sea turtle L. olivacea and several mammalian species.
Lines 707-709
Authors: “Therefore, the onset of gonadal sex differentiation, just as the formation of the gonadal ridges, is another event synchronized with somatic development in vertebrates.”
My comment: “synchronized with the somatic development” is too broad and weak; in my opinion, a more precise correlation could be “synchronized with the limb buds.”
Lines 750-762:
My comment: This paragraph mostly repeats parts of the text in the Results section. I consider that it can be deleted or reduced considerably.
Lines 774-778:
The T-shaped domain refers to the Sox9 positive mesothelial cells of the dorsal mesentery, which are displaced towards the forming genital ridges. Since using traditional histology, Sox9+ cells look like other mesothelial cells, it would be necessary to detect the expression of such transcription factor using immunohistochemical techniques to discard the presence of the T-shaped domain in geckos.
Author Response
Reviewer 1:
Comments to the Introduction
Lines 51-56
Authors: “However, this is an oversimplification. A large portion of the ovary called the stroma, also develops from the medulla, while the cortex in the testis forms the epithelium covering the male gonad. Within the ovarian cortex, germline nests develop, which then fragment into ovarian follicles. In the testicular medulla, on the other hand, testis cords develop and transform into seminiferous tubules.
My comment:
Witchi’s classical studies in amphibia did not consider the stromal tissue in the ovary. However, in developing gonads of Amniota vertebrates, stromal tissues are mostly formed by mesenchymal cells, fibroblasts, and blood vessels.
Response: line 65 - Indeed, the sentence was misleading and we decided to remove the part “, called the stroma,” to make the sentence correct.
Comments to Material and Methods
Figure 2
My comment:
The Balbiani body (Bb) of germ cells is shown only in Fig. 2G. It would be useful to show it in Fig. 2H also. I suggest the authors mention the organelle composition of the (Bb).
Response: We added an insert to Fig. 2E with a magnified view of the germ cell. We labeled the Balbiani body on this insert and in Fig. 2H. We included the corresponding text in the figure description - line 240.
Figure 3
Figs. 3C and 3D
The authors fail to mention that E-cadherin positive or negative cells correspond to epithelial or stromal gonadal tissues. Laminin is indicated only in the cortex of Fig. 3E (arrows); however, the thinner positive mark around the medullary cords is not mentioned. This point is important as the epithelial origin of the medullary cords from the surface epithelium was found in the genital ridges of turtles and mammals.
Response: line 293 – We added: “E-cadherin positive or negative cells correspond to epithelial or stromal gonadal tissues, respectively”. We have also added a new photo, Fig. 3F, showing laminin around the forming medullary cord.
Figure 4
Fig. 4D
Proliferating cells are not indicated.
Response: We added arrowheads indicating PCNA-positive sites.
Figs. 4E and 4F
My comments:
Although E-cadherin-positive epithelial tissues are evident in 4E, the surrounding laminin-positive medullary cords in 4F are not correctly mentioned. The authors mention that “laminin strongly accumulates in the stroma.” This point is important because the basal lamina, which segregates epithelial from stromal tissues, is important in establishing the undifferentiated (bipotential) gonads before their dimorphic sexual morphogenesis as testes or ovaries.
Response: Our immunostaining of laminin showed that positive result in the whole stroma, which suggests that laminins can be present in the whole stroma, not only in the basal lamina at the border between the epithelium and stroma. We suggest to potin the laminin-positive stroma with arrows as it is now.
Figure 5
Fig. 5E lines 369-361.
Authors: “Laminin immunostaining showed that gonadal stroma was dispersed in differentiating testes and did not form one layer between the cortex and medulla as in the ovaries.”
My comment: The protein laminin is produced by epithelial cells and then deposited on the surface facing the stromal tissue.
Response: Indeed we observed that the whole stroma had been positive for laminin. Thus, this immunostaining appeared useful to trace formation of the stroma and internal gonadal structure. We mentioned it in line 287: Immunostaining of laminin demonstrates the presence of this protein in the mesen-chyme (gonadal stroma; arrows on Fig. 3E), forming a layer between the gonadal cor-tex and the medullary cords.
Figure 7
Lines 430-433.
Authors: “Moreover, the gonadal ridges in the leopard gecko also had a form of two bulges composed of: 1) a primitive cortex derived from a thickened coelomic epithelium and containing germ cells; and 2) a primitive medulla as an accumulation of mesenchymal cells located inside the gonadal ridge (Fig. 7D; gr, gonadal ridges).”
My comment: The “accumulation of mesenchymal cells” in the genital ridges might be incorrect because their differentiation state cannot be yet ascertained.
Response: In our opinion, in Fig. 7D, there is a distinct cluster of mesenchymal cells on one side of the basement membrane, indicating their accumulation. We modified the sentence, however, we propose to keep the word 'accumulation' (line 457).
Lines 435-437
“The primitive medulla in both species was a cluster of somatic cells continuous with the mesonephric mesenchyme, which suggests that the gonadal medulla was formed by the aggregation of mesenchymal cells.”
My comment: Unfortunately, I think there is confusion in using the term “medulla.” As a spatial gonadal site, the gonadal medulla contains the epithelial medullary cords and their surrounding stromal tissue formed by mesenchymal cells, fibroblasts, and blood vessels. If the medullary cords are formed by “aggregation” of mesenchymal cells, one must assume that, in contrast with mammals and turtles, a mesenchymal to epithelial transition occurs in geckos.
Response: We fully agree with the reviewer's suggestions. We have rephrased the text to clearly emphasize this issue.
Figure 9
Lines 512.513
“The zone of E-cadherin–stromal cells contained laminin, indicating their mesenchymal nature (Fig. 9H).”
However, the authors again assume a Mesenchymal-epithelial transition may occur in geckos. However, Figure 9G shows a clear continuity of E-cadherin-positive cells among the epithelial cortex and a medullary cord, suggesting their common origin.
Response: We fully agree with the reviewer's suggestions. We have rephrased the text to clearly emphasize this issue.
Figure 10
Lines 583-584: “Yolk droplets were present when the germ cells invaded the gonadal ridges at S29, but the yolk disappeared soon after as in other geckos.”
My comment: Although this is an interesting physiological detail, no pictures are shown.
Response: we added the insert to Fig. 11C showing details of the germ cell structure.
Figure 11
Figs.11M and 11N. Although the authors mention the presence of “dispersed streams of E-cadherin+ somatic cells,” in my opinion, they correspond to medullary cords.
Response: In fact, these may be medullary cords, but their shape and weak E-cadherin signal suggest that they are not as advanced as those in the leopard gecko. We have added an explanation to the text.
Authors: Lines 589-581: “Immunostaining showed the presence of a cluster of E-cadherin+ cells in the medulla. This cluster extended to the mesonephros (Fig. 11E). A layer of E-cadherin–stromal cells (black arrows on Fig. 11E) was present between the medulla and the cortex, both positive for E-cadherin.
My comment: In this paragraph, the authors clearly distinguish between the two kinds of tissues in the medulla of developing gonads: (1) E-cadherin-positive medullary cords and (2) E-cadherin-negative stromal cells. I suggest replacing +/- with positive/negative.
Response: We used “E-cadherin-positive” and “E-cadherin-negative” as the reviewer suggested. And we added information that these structures are rudimentary medullary cords.
Discussion
Lines 661-662:
Authors: “This indicates that at such an early stage, the initiation of gonadal development is synchronized with somatic development.”
My comment: This is an interesting point; however, the precise correlation is between the paired buds and the establishment of the genital ridges rather than “with somatic development.”
Response: We clarified that it pertains to synchronization with the development of limb buds rather than overall somatic development (line 694).
Lines 686-689
Authors: “In this study, we have demonstrated that in geckos, medullary cords are connected to the cortex, indicating that they form as an outgrowth of the cortex, similar to what has been shown 688 in T. scripta [10]. This indicates the origin of the medullary cords from the gonadal cortex.”
My comment: This paragraph supports my comments on the author’s figures interpretation in their Introduction and Results. The epithelial origin of medullary cords from the mesothelial cells has also been shown in other amniotes as the sea turtle L. olivacea and several mammalian species.
Response: We fully agree with the reviewer's comments. We have rephrased the text to clearly emphasize this issue.
Lines 707-709
Authors: “Therefore, the onset of gonadal sex differentiation, just as the formation of the gonadal ridges, is another event synchronized with somatic development in vertebrates.”
My comment: “synchronized with the somatic development” is too broad and weak; in my opinion, a more precise correlation could be “synchronized with the limb buds.”
Response: We corrected this sentence as the reviewer suggested (line 765).
Lines 750-762:
My comment: This paragraph mostly repeats parts of the text in the Results section. I consider that it can be deleted or reduced considerably.
Response: We have rephrased the discussion to avoid repetitions.
Lines 774-778:
The T-shaped domain refers to the Sox9 positive mesothelial cells of the dorsal mesentery, which are displaced towards the forming genital ridges. Since using traditional histology, Sox9+ cells look like other mesothelial cells, it would be necessary to detect the expression of such transcription factor using immunohistochemical techniques to discard the presence of the T-shaped domain in geckos.
Response: We have rephrased this section to refer to the T-domain and indicate that further studies using SOX9 localization should clarify whether the T-domain is present in this region (833).

Reviewer 2 Report
Comments and Suggestions for Authors
Revision
Manuscript: Histological analysis of gonadal ridge development and sex differentiation of gonads in three gecko species
This manuscript by Rams-Pociecha et. al., described the morphology of the gonadal ridge and the histological changes during gonadal sex differentiation for both ovary and testis from three species belonging to the Gekko infraorder: Correlophus ciliates, Eublepharis macularius and Lepidodactylus lugubri. The manuscript is well written and provide useful information about the gonadal sex differentiation of three species that have different systems of sex determination. One species, in particular, can be used as experimental model for future studies. Therefore, this study is important since provide morphological basis to support further studies aiming to investigate genes involved in the male or female sex differentiation. Although interesting, this reviewer found the manuscript too long, sometimes repetitive, and legend of the figures should be carefully revised.
I will provide some corrections/suggestions, but I stongly recommend that authors revise carefully the legend of all figures.
Suggestions/Comments:
Line 47: somatic cells.
Line 61: please provide examples.
Figure 1B: please provide detail description of the experimental design and periods of sampling.
What the different colours in the lines represent?
Lines 150-162: Authors should mention Figure 1B and explain better the experimental design.
Are the antibodies used in this study specific? Did the authors perform a positive control?
Please provide details of the histomorphometric analysis, for example, number of sections evaluated for measuring cell numbers and thickness of the cortex.
Provide better and more details regarding the statistical analysis. Please provide the statistic test used and p value considered.
Figure 2C: vs is not indicated. It is mentioned in the legend but not indicated in the figure.
Figure 2E: cm is not indicated. It is mentioned in the legend but not indicated in the figure.
Figure 2H: co’ is not indicated in the figure.
Authors mentioned Balbiani body, however, magnification is not higher enough to identify the Balbiani’s body. Please provide an inset.
Lines 257-259: Authors should use an arrow in the figure to indicate this information.
Please change the colour of all arrows/arrowheads to white. This would make it clear the identification and visualization.
Figure 3C: There is an asterisk in the Figure 3C, but no mention in the legend.
Line 292: Fig 4E
Fig. 4E: There is an asterisk in the Figure 4E, but no mention in the legend.
Figure 7E: please indicate the undifferentiated gonads.
Figure 8: These figures A and B look very similar. How do authors differentiate testis from ovary. Please explain.
Figure 9: please provide the stage for D, F and H.
Suggestion 1: Result section is too long. Authors should consider rewriting the results in order to avoid repetitive description. For example: make a description comparing the three species, especially for the undifferentiated gonad. This could be a strategy to reduce the length of the text and avoid repetitions.
Suggestion 2: Authors should make a comparative scheme or Figure grouping the many morphological differences for each species in the undifferentiated stage, testis differentiation and ovary differentiation.
Author Response
Reviewer 2:
Manuscript: Histological analysis of gonadal ridge development and sex differentiation of gonads in three gecko species
This manuscript by Rams-Pociecha et. al., described the morphology of the gonadal ridge and the histological changes during gonadal sex differentiation for both ovary and testis from three species belonging to the Gekko infraorder: Correlophus ciliates, Eublepharis macularius and Lepidodactylus lugubri. The manuscript is well written and provide useful information about the gonadal sex differentiation of three species that have different systems of sex determination. One species, in particular, can be used as experimental model for future studies. Therefore, this study is important since provide morphological basis to support further studies aiming to investigate genes involved in the male or female sex differentiation. Although interesting, this reviewer found the manuscript too long, sometimes repetitive, and legend of the figures should be carefully revised.
I will provide some corrections/suggestions, but I strongly recommend that authors revise carefully the legend of all figures.
Response: Thank you for this comment on our manuscript. We have once again reviewed the text along with the figure legends to correct all errors and oversights.
Suggestions/Comments:
Line 47: somatic cells.
Response: We added “somatic” as the reviewer suggested.
Line 61: please provide examples.
Response: We added information in line 74 that testes undergo an initial ovarian phase in zebrafish and several Rana species.
Figure 1B: please provide detail description of the experimental design and periods of sampling.
Response: We have provided additional information on sampling in the section of Materials and Methods (line 167).
What the different colours in the lines represent?
Response: In the figure legend, we added a description indicating that “the red line indicates the phase of undifferentiated gonads, while the green line represents the phase of differentiated gonads”.
Lines 150-162: Authors should mention Figure 1B and explain better the experimental design.
Response: At the beginning of the Materials and Methods section, we added text regarding the study design, and we cited Fig. 1B (lines 167).
Are the antibodies used in this study specific? Did the authors perform a positive control?
Response: Prior to antibody procurement, we verified the protein sequences of our targets against the peptide sequences used in antibody production to ensure positive study outcomes. The application of anti-E-cadherin antibodies stained epithelial cells, while anti-laminin antibodies stained the stroma, confirming their specificity.
Please provide details of the histomorphometric analysis, for example, number of sections evaluated for measuring cell numbers and thickness of the cortex.
Response: We have added a description to the Materials and Methods section since the details of the histometric method were not clear. The results of these analyses are presented in Suppl. Table 2.
Provide better and more details regarding the statistical analysis. Please provide the statistic test used and p value considered.
Response: As mentioned above.
Figure 2C: vs is not indicated. It is mentioned in the legend but not indicated in the figure.
Response: We added vs in Fig. 2C, as suggested.
Figure 2E: cm is not indicated. It is mentioned in the legend but not indicated in the figure.
Response: In the figure legend, we incorrectly typed cm. It should have been co instead of cm, and we have already corrected this in the revised version of the manuscript (line 239).
Figure 2H: co’ is not indicated in the figure.
Response: we used co1, co2 and co3 in Fig. 2.
Authors mentioned Balbiani body, however, magnification is not higher enough to identify the Balbiani’s body. Please provide an inset.
Response: We have added inserts with details of the germ cell structure for each species.
Lines 257-259: Authors should use an arrow in the figure to indicate this information.
Response: We added this information in lines 277 and 281 to clarify that clusters of E-cadherin-negative cells are indicated by arrows.
Please change the colour of all arrows/arrowheads to white. This would make it clear the identification and visualization.
Response: We used arrows in different colors on the figures to indicate different cells. For example, germ cells in the cortex and germ cells in the medulla were indicated by arrows in different colors. We suggest keeping this color differentiation as it is. If the reviewers find them less visible, we propose increasing the size of the arrows.
Figure 3C: There is an asterisk in the Figure 3C, but no mention in the legend.
Response: The asterisk is mentioned in the legend and indicates a primordium of efferent ducts (asterisk) – line 296.
Line 292: Fig 4E
Response: This part of the text specifically pertain to the ovary, so we do not mention Fig. 4E in this context.
Fig. 4E: There is an asterisk in the Figure 4E, but no mention in the legend.
Response: We added the explanation in line 362.
Figure 7E: please indicate the undifferentiated gonads.
Response: We added information to the legend that the arrows indicate the undifferentiated gonads – line 494.
Figure 8: These figures A and B look very similar. How do authors differentiate testis from ovary. Please explain.
Response: We modified the description of Figure 8 to clarify that the presence of a thick cortex containing numerous germ cells indicates ovarian differentiation.
Figure 9: please provide the stage for D, F and H.
Response: We have provided stages in Fig. 9.
Suggestion 1: Result section is too long. Authors should consider rewriting the results in order to avoid repetitive description. For example: make a description comparing the three species, especially for the undifferentiated gonad. This could be a strategy to reduce the length of the text and avoid repetitions.
Response: We have rephrased the discussion to avoid repetitions.
Suggestion 2: Authors should make a comparative scheme or Figure grouping the many morphological differences for each species in the undifferentiated stage, testis differentiation and ovary differentiation.
Response: We have added a new figure 12.

Reviewer 3 Report
Comments and Suggestions for Authors
This manuscript written by Rams-Pociecha et al. concerns gonadal development in geckos. The reptiles are interesting animals to study the evolution of sex determination because their sex determining mechanisms are divergent. The authors use three different gecko species, each of which has either a sex chromosome-dependent, temperature-dependent or parthenogenetic sex determination system. Gonadal development has not been explored in geckos except for a few specific species. It would be nice to investigate new species that are used in this work. The authors demonstrated distinct and conserved processes in testicular and ovarian differentiation around the sex-determining stage by histological analyses. Hematoxylin and picroaniline (HP) staining provides evident gonadal morphology. However, molecular characterization, particularly the expression of cell-type specific markers, are too poor to warrant publication in Biology. This topic may not be suitable for broad readership of this journal, but a journal specialized for developmental biology or reproductive biology.
Specific comments:
1. E-cadherin is used as a marker for the developing cortex and medulla in this study. However, E-cadherin immunostaining is less informative than other markers because it is expressed in germ cells and all gonadal somatic cells (except for stroma cells) in the three gecko species. I wonder if specific markers for cortical or medullary cells shown in other animals, e.g. Sox9, Foxl2, Nr5a1, Wt1 etc, are available in geckos.
2. E-cadherin expression is present not only in the plasma membrane but in the nuclei. I wonder if nuclear E-cadherin has a particular role.
3. It is known that the thickening coelomic epithelium, or the cortex, is proliferative in other animals. In the crested gecko, the coelomic epithelium (Fig. 3A) and the ovarian cortex (Fig. 4D) are PCNA-positive, while the testicular cortex is not (Fig. 5C). Only a few interior cells are proliferative (line 366). I wonder how the testis develops with such a few proliferative cells. In contrast, the ovary is less proliferative than the testis in the leopard gecko (Fig. 8A,B). How is the difference between the two gecko species explained?
4. The demarcation lines or circles indicate the cortex-medulla boundaries or the outline of testis cords. Although the authors claim the formation of a basement membrane at the boundaries or testis cord on HP staining sections, molecular evidence is poor. Laminin can be used as a basement membrane maker. However, laminin expression does not always demarcate the boundaries. Are there any markers that can verify the basement membrane formation in geckos?
5. The authors mention the loss of germ cells (line 543), but it is not addressed. Is it because apoptotic germ cell death? If so, when does it occur? Immunostaining for apoptosis could be performed.
6. There are multiple images at the same stages; four images at S35 in Fig. 4G-J, two images at S34 in Fig. 5G and H, two images at S34 in Fig. 6A and B, and two images at S30 in Fig. 7C and D. Are they a representative image from different individuals or from the same animal but different section levels along with the anteroposterior axis?
Other comments:
Fig. 5N and O look like different staining or microscopy than the others.
In Fig. 7A and B, germ cells look unusually large compared to the size of the genital ridge. Are they really a single germ cell?
What is the sex chromosomal complement of the intersex gonads shown in Fig. 6, ZZ or ZW?
There are large and small asterisks in figures. They are confusing. Different colours can be used.
Several abbreviations are missing from figure legends. For example, ‘c’ and ‘cc’ in Fig. 5A and B
Fig. 4J lacks details in the figure legend.
Line 290.What is ‘lover signal’? Is it ‘lower’?
Line 308. There is no red arrowhead in Fig. 4G-M.
Line 359. The number of germ cells in the cortex decreases.
Line 368. Strong E-cadherin signal is observed not only in the testis cord but in the dorsal cortex.
Line 380. There is no red arrowhead in Fig. 5H.
Line 386. Fig. 5F has no arrow.
Comments on the Quality of English LanguageThe manuscript is well written in English. There are only a few typos or grammatical errors that can be corrected easily.
Author Response
Reviewer 3:
This manuscript written by Rams-Pociecha et al. concerns gonadal development in geckos. The reptiles are interesting animals to study the evolution of sex determination because their sex determining mechanisms are divergent. The authors use three different gecko species, each of which has either a sex chromosome-dependent, temperature-dependent or parthenogenetic sex determination system. Gonadal development has not been explored in geckos except for a few specific species. It would be nice to investigate new species that are used in this work. The authors demonstrated distinct and conserved processes in testicular and ovarian differentiation around the sex-determining stage by histological analyses. Hematoxylin and picroaniline (HP) staining provides evident gonadal morphology. However, molecular characterization, particularly the expression of cell-type specific markers, are too poor to warrant publication in Biology. This topic may not be suitable for broad readership of this journal, but a journal specialized for developmental biology or reproductive biology.
Specific comments:
- E-cadherin is used as a marker for the developing cortex and medulla in this study. However, E-cadherin immunostaining is less informative than other markers because it is expressed in germ cells and all gonadal somatic cells (except for stroma cells) in the three gecko species. I wonder if specific markers for cortical or medullary cells shown in other animals, e.g. Sox9, Foxl2, Nr5a1, Wt1 etc, are available in geckos.
Response: Finding antibodies that will specifically work across three gecko species is a challenge. From our experience, the expression of genes involved in sex determination, such as Sox9 and Nr5a1, is often not suitable for tracking changes in the developing gonad structure, as their expression extends to other structures within the gonad. In previous studies, we used E-cadherin because it effectively labels epithelial structures, distinguishing them from the stroma. This allows for a clear visualization of gonadal structure.
- E-cadherin expression is present not only in the plasma membrane but in the nuclei. I wonder if nuclear E-cadherin has a particular role.
Response: E-cadherin shows very high expression. Indeed, in Fig. 5D, a signal is visible in the nucleus. This is not typical. It may be due to excessive immunohistochemical staining.
- It is known that the thickening coelomic epithelium, or the cortex, is proliferative in other animals. In the crested gecko, the coelomic epithelium (Fig. 3A) and the ovarian cortex (Fig. 4D) are PCNA-positive, while the testicular cortex is not (Fig. 5C). Only a few interior cells are proliferative (line 366). I wonder how the testis develops with such a few proliferative cells. In contrast, the ovary is less proliferative than the testis in the leopard gecko (Fig. 8A,B). How is the difference between the two gecko species explained?
Likely, the epithelial cells surrounding the testis also divide because, with the growth of the nucleus, its surface area increases, but these divisions are not intensive. Fig. 8B shows that certain cells are PCNA-positive. However, cells of the superficial epithelium of the differentiating ovaries proliferate at a significant level, which results in a thick cortex.
- The demarcation lines or circles indicate the cortex-medulla boundaries or the outline of testis cords. Although the authors claim the formation of a basement membrane at the boundaries or testis cord on HP staining sections, molecular evidence is poor. Laminin can be used as a basement membrane maker. However, laminin expression does not always demarcate the boundaries. Are there any markers that can verify the basement membrane formation in geckos?
Response: In our case, histological staining best depicted the course of the basement membrane. We decided to use anti-laminin antibodies to demonstrate this using the IHC method. It turned out that anti-laminin antibodies stained the entire stroma. This effect proved useful for visualizing the gonadal stroma. Perhaps laminins are deposited throughout the developing gonadal stroma. We obtained a similar effect using these antibodies in immunofluorescence on frog gonads.
- The authors mention the loss of germ cells (line 543), but it is not addressed. Is it because apoptotic germ cell death? If so, when does it occur? Immunostaining for apoptosis could be performed.
Response: This study did not allow us to understand the mechanism of germ cell loss. Its aime was to describe the structural changes in the developing gonads. We plan to conduct further research using the TUNEL method, which will help determine the involvement of apoptosis in sexual differentiation of the gonads. However, this requires extensive research and analysis and will be the subject of a subsequent publication.
- There are multiple images at the same stages; four images at S35 in Fig. 4G-J, two images at S34 in Fig. 5G and H, two images at S34 in Fig. 6A and B, and two images at S30 in Fig. 7C and D. Are they a representative image from different individuals or from the same animal but different section levels along with the anteroposterior axis?
Response: In some figures, we included gonads from different individuals. This illustrates that individuals vary in gonadal shape. These are gonads from different individuals. In the figure legend, it is stated, for example, that Fig. 4G corresponds to stage S35 on day D17, while Fig. 4J represents the same stage (S35) but a slightly later developmental day, D19. Figs. 5G and H depict exactly the same stage and day, but two different individuals showing slightly different gonadal shapes. Fig. 6. Two individuals had intersexual gonads, and in this figure, we presented the gonads of these two individuals. We have added this information in line 445. Figs. 7C and D come from two different individuals.
Other comments:
Fig. 5N and O look like different staining or microscopy than the others.
Response: Photographs in figures 5N and 5O come from a separate series of stainings. The intense staining of cell nuclei resulted from the use of fresh hematoxylin. All testes from this stage were stained in this way, so we cannot replace them with other images. It is exactly the same staining as the others, only the tissues are more intensely stained.
In Fig. 7A and B, germ cells look unusually large compared to the size of the genital ridge. Are they really a single germ cell?
Response: Indeed, these are individual germ cells. Their characteristic feature is their large size.
What is the sex chromosomal complement of the intersex gonads shown in Fig. 6, ZZ or ZW?
Response: Due to the absence of specific primers, we are currently unable to determine the genetic genders of these individuals.
There are large and small asterisks in figures. They are confusing. Different colours can be used.
Response: We have modified the figures so that the asterisks are of uniform size.
Several abbreviations are missing from figure legends. For example, ‘c’ and ‘cc’ in Fig. 5A and B
Response: We explained cc abbreviation in the Fig. 5. We changed c for cc (line 421).
Fig. 4J lacks details in the figure legend.
Response: We added additional detail in the figure legands (line 366).
Line 290.What is ‘lover signal’? Is it ‘lower’?
Response: We have corrected that word to 'lower'.
Line 308. There is no red arrowhead in Fig. 4G-M.
Response: we added red arrowheads to the modified version of our manuscript.
Line 359. The number of germ cells in the cortex decreases.
Response: True, as we stated in the sentence.
Line 368. Strong E-cadherin signal is observed not only in the testis cord but in the dorsal cortex.
Response: As E-cadherin is always present in the epithelium, we omitted the information that it is in the surface epithelium of the gonads as obvious, to make the message clearer. It was crucial here to demonstrate the testis cords.
Line 380. There is no red arrowhead in Fig. 5H.
Response: we added the red arrowhead as suggested.
Line 386. Fig. 5F has no arrow.
Response: we added „mg” in Fig. 5F and mentioned it in line 409.

Round 2
Reviewer 2 Report
Comments and Suggestions for Authors
Authors have responded satisfactorily to all my queries. They've revised the manuscript as per my suggestions and made a beautiful comparative scheme (Figure 12).
Author Response
We would like to express our gratitude to the reviewer for his/her time and insightful feedback on our article. I am pleased to learn that the revisions we made have met his/her satisfaction. The constructive comments provided have been invaluable in improving the manuscript.
Reviewer 3 Report
Comments and Suggestions for Authors
The authors have improved the manuscript. Considering that the manuscript is submitted to the special issue ‘Mechanisms of Sex Determination and Gonadal Development’ of this journal, the current topic is suitable for specific readers. The addition of figure 12 would help to summarize the results.
However, I am still concerned with the lack of cell-type-specific markers. E-cadherin is used as an epithelial marker in the current study, but it would not tell us whether testis cords or ovarian medullary cords are derived from the coelomic epithelium cells. As reviewer 1 suggested, E-cadherin positive cells could arise from the mesenchymal cells by a mesenchymal-epithelial transition.
Although the authors excuse the unavailability of suitable antibodies, e.g. Sox9 and Nr5a1, for tracking cells in geckos, in situ hybridization should be good enough to label specific cell types. In fact, the authors refer to a Sox9 in situ hybridization study in the Leopard gecko (Ref 19). The combination of in situ hybridization and antibody staining is technically possible in many cases. I wonder whether the combination of E-cadherin immunostaining and in situ hybridization of either Sox9, Foxl2, or Nr5a1 is feasible. This will be more informative to reveal the molecular basis of sex differentiation of gonads in geckos.
Author Response
We would like to express our sincere appreciation for reviewer's time and thoughtful feedback, which has contributed significantly to the improvement of our article. Investigating the origin of testis cords from the cortex, as well as the overall gonadal medulla origin, poses considerable challenges. Histological cross-sections and immunostaining analyses provide clues suggesting whether the medulla originates from the coelomic epithelium or the mesonephros. However, such analyses can only offer suggestive evidence. For instance, Figures 3A, 4E, 7H, I, and 9G illustrate the presence of E-cadherin-positive cells in the medullary cords that are continuous with the cortex.
We conducted various histological and immunohistochemical analyses that could shed light on the formation of testis cords from the cortex.
Direct evidence of the origin of medullary cells from the coelomic epithelium or the mesonephros, may be provided by cell tracing analysis, which tracks migrating cells under in vitro conditions. Unfortunately, we lack live gecko embryo material for in vitro studies. Similarly, we do not possess material suitable for in situ hybridization analysis. The examination of Sox9, Foxl2, and Nr5a1 gene expression was not within the scope of this article, as our focus was on presenting structural changes occurring in developing gecko gonads. The absence of live gecko embryo cultures and suitable material for in situ hybridization analysis prevents us from conducting such experiments.
The molecular mechanisms of sex determination in geckos, including the analysis of Sox9, Foxl2, and Nr5a1 gene expression, are subjects more appropriate for a separate article. Our goal in this study was to present structural changes in developing gecko gonads, laying a foundation for further research, particularly in sex determination mechanisms.
Once again, thank you for your invaluable feedback and suggestions.
Round 3
Reviewer 3 Report
Comments and Suggestions for Authors
I have no further concern. The manuscript can be accepted.